# Towards Understanding Adam Convergence on Highly Degenerate Polynomials

**Zhiwei Bai** [* 1 2]   **Jiajie Zhao** [* 1 2]   **Zhangchen Zhou** [1 2]   **Zhi-Qin John Xu** [1 2 3]   **Yaoyu Zhang** [1 2]

## Abstract

Adam is a widely used optimization algorithm in deep learning, yet the specific class of objective functions where it exhibits inherent advantages remains underexplored. Unlike prior studies requiring external schedulers and $\beta_2$ near 1 for convergence, this work investigates the "natural" auto-convergence properties of Adam. We identify a class of highly degenerate polynomials where Adam converges automatically without additional schedulers. Specifically, we derive theoretical conditions for local asymptotic stability on degenerate polynomials and demonstrate strong alignment between theoretical bounds and experimental results. We prove that Adam achieves local linear convergence on these degenerate functions, significantly outperforming the sub-linear convergence of Gradient Descent and Momentum. This acceleration stems from a decoupling mechanism between the second moment $v_t$ and squared gradient $g_t^2$, which exponentially amplifies the effective learning rate. Finally, we characterize Adam's hyperparameter phase diagram, identifying three distinct behavioral regimes: stable convergence, spikes, and SignGD-like oscillation.

## 1. Introduction

Adam (Kingma & Ba, 2014) is widely used in deep learning optimization, yet theoretical understanding of which problem types inherently favor Adam over gradient descent (GD) and momentum methods remains limited. In particular, Adam's convergence itself has been a persistent challenge: Reddi et al. (2018) demonstrated that Adam can fail to converge even in simple convex settings. Numerous studies

---
[*]Equal contribution [1]Institute of Natural Sciences, MOE-LSC, Shanghai Jiao Tong University [2]School of Mathematical Sciences, Shanghai Jiao Tong University [3]Shanghai Seres Information Technology Co., Ltd, Shanghai 200040, China. Correspondence to: Zhi-Qin John Xu <xuzhiqin@sjtu.edu.cn>, Yaoyu Zhang <zhyy.sjtu@sjtu.edu.cn>.

*Proceedings of the 43rd International Conference on Machine Learning*, Seoul, South Korea. PMLR 306, 2026. Copyright 2026 by the author(s).

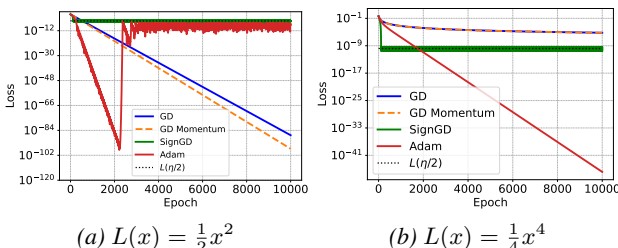

*Figure 1.* Convergence behavior differs between strongly convex and degenerate polynomials. Adam uses $\beta_1 = 0.9, \beta_2 = 0.99$.

have since focused on the convergence properties of adaptive gradient methods (Chen et al., 2019; Li & Orabona, 2019; Xie et al., 2020; Défossez et al., 2022; Da Silva & Gazeau, 2020; Shi et al., 2021; Zou et al., 2019; Zhou et al., 2024; Zhang et al., 2022a). Among these, Zhang et al. (2022a) showed that with decreasing learning rate scheduler $\eta/\sqrt{T}$, Adam can converge at a rate of $O(\log(T)/\sqrt{T})$ when $\beta_2$ is sufficiently close to 1 and $\beta_1 < \sqrt{\beta_2}$. Nevertheless, the auto-convergence properties of Adam without external schedulers remain underexplored—specifically, **the function classes where Adam "naturally" exhibits convergence advantages over GD are still poorly understood.**

This work investigates Adam's convergence behavior on highly degenerate polynomials. Our motivation comes from the observation in Fig. 1: on the strongly convex function $L(x) = \frac{1}{2}x^2$, constant-stepsize Adam initially exhibits linear convergence similar to gradient descent and momentum, but eventually suffers from loss spikes as studied in Cohen et al. (2023); Bai et al. (2025). In contrast, on the degenerate function $L(x) = \frac{1}{4}x^4$, Adam achieves stable *linear* convergence without instabilities, while GD and Momentum methods degrade to *sublinear* convergence. This distinction is particularly crucial because, although classical optimization theory focuses on strongly convex cases, extensive research shows that deep learning loss landscapes contain many highly degenerate directions (Sagun et al., 2017; Zhang et al., 2021; 2022b; Bai et al., 2024; Fukumizu et al., 2019; Simsek et al., 2021; Zhang et al., 2025).

Focusing on degenerate objectives $L(x) = \frac{1}{k}x^k$ ($k \geq 4$, even), we derive theoretical conditions for local asymptotic stability with strong empirical alignment. We prove

that Adam achieves linear convergence on these functions, significantly outperforming the sublinear rates of GD and Momentum. We explicitly characterize this acceleration arises from a decoupling mechanism between the second moment estimate $v_t$ and the squared gradient $g_t^2$, which exponentially amplifies the effective learning rate. We also characterize Adam's hyperparameter phase diagram of different behaviors, identifying three distinct regimes: stable convergence, spikes, and SignGD-like oscillation.

Our main contributions are:

(i) We identify a class of highly degenerate polynomials where Adam converges without learning rate schedulers, deriving local convergence conditions over the full hyperparameter domain $[0, 1)$ that generalize prior results requiring $\beta_2$ close to 1 (Zhang et al., 2022a).

(ii) We prove Adam achieves local linear convergence on degenerate functions, outperforming GD's sublinear rate through a decoupling mechanism between $v_t$ and $g_t^2$ that exponentially amplifies the effective learning rate, distinct from prior SignGD-based acceleration mechanism studied in Kunstner et al. (2023; 2024).

(iii) We systematically characterize Adam's phase diagram across hyperparameters, identifying three regimes—stable convergence, spikes, and SignGD-like oscillation—that explain empirical behaviors observed in prior works (Ma et al., 2022; Bai et al., 2025).

## 2. Related Works

**Convergence Analysis of Adaptive Methods.** Extensive research has analyzed the convergence of adaptive methods. Several works identify failure modes: Reddi et al. (2018) demonstrated Adam can fail in simple convex settings, and recent work reveals its non-convergence on quadratics under asymmetric distribution (Dereich et al., 2025a), while Da Silva & Gazeau (2020); Bock & Weiß (2019) showed RMSProp and Adam exhibit limit cycles on quadratic functions, and Ma et al. (2022); Bai et al. (2025) observed spiking, oscillation, or divergence depending on hyperparameters. Under appropriate hyperparameter selection, adaptive methods can achieve convergence rates of $O(\log T/\sqrt{T})$, typically requiring learning rate decay (Chen et al., 2019; Zou et al., 2019; Ward et al., 2020; Shen et al., 2023; Défossez et al., 2022; Zhang et al., 2022a; Da Silva & Gazeau, 2020; Barakat & Bianchi, 2021; Dereich & Jentzen, 2024). Recent continuous-time ODE approximations and rigorous a priori bounds have further refined these convergence rates and global stability analyses (Dereich et al., 2025c;b; 2026). In stochastic settings, Dereich et al. (2024) showed adaptive methods cannot converge without vanishing learning rates. In contrast, our work demonstrates that Adam achieves linear convergence on deterministic degen-

erate functions *without* learning rate decay.

**Degeneracy in Deep Learning Loss Landscapes.** Deep learning loss landscapes exhibit pervasive high-order degeneracy, with empirical Hessian analysis revealing eigenvalues concentrated near zero (Sagun et al., 2017). Several studies investigate this degeneracy through the symmetry of neural network parameterization (Fukumizu et al., 2019; Simsek et al., 2021; Zhang et al., 2025). The Embedding Principle further shows that critical points of smaller networks embed into larger networks as high-dimensional degenerate manifolds (Zhang et al., 2021; 2022b; Bai et al., 2024).

**Adam Outperforms Gradient Descent.** Understanding why Adam outperforms GD has been extensively studied. While heavy-tailed stochastic noise in language tasks was initially proposed (Zhang et al., 2020), Chen et al. (2021); Kunstner et al. (2023) showed the performance gap persists even in deterministic full-batch settings. Proposed mechanisms include Hessian block heterogeneity (Zhang et al., 2024), gradient heterogeneity (Tomihari & Sato, 2025), coordinate-wise $\ell_\infty$ geometry exploitation (Xie et al., 2024), and resilience to heavy-tailed class imbalance (Kunstner et al., 2024). Recently, Davis et al. (2025) proved that standard gradient descent yields sublinear convergence on highly degenerate functions, whereas adaptive step sizes via additional Polyak's rule achieve linear convergence. Compared to this, our work focuses on Adam's *inherent adaptation mechanism* via second-order moments, demonstrating linear convergence without *any* external modifications.

To further clarify the distinction between our theoretical framework and existing literature on Adam's advantages over SGD, we summarize the key mechanisms and their structural assumptions in Table 1. It is important to note that these mechanisms are not mutually exclusive; multiple factors likely operate simultaneously during practical training. Our analysis isolates a fundamental, landscape-driven factor that has been largely overlooked in previous theoretical treatments.

## 3. Preliminaries and Problem Setup

### 3.1. The Model Problem: High-Order Degeneracy

In this work, we consider the local behavior around a degenerate minimum $x^* = 0$ where the first $k - 1$ derivatives vanish. In particular, we study the prototype function:

$$L(x) = \frac{1}{k}x^k, \quad \text{where } k \geq 4 \text{ is an even integer.} \tag{1}$$

The gradient is $\nabla L(x) := L'(x) = x^{k-1}$ and the Hessian $\nabla^2 L(x) := L''(x) = (k-1)x^{k-2}$ vanishes as $x \to 0$.

*Table 1.* Comparison of theoretical mechanisms explaining Adam's advantages over SGD. While prior works often rely on specific assumptions regarding noise, geometry, or data distribution, our proposed mechanism operates solely on loss landscape degeneracy—a ubiquitous structural property in deep neural networks that does not require stochasticity.

| Work | Proposed Mechanism | Key Assumption | Requires Stochasticity |
|------|--------------------|-----------------|------------------------|
| Zhang et al. (2020) | Heavy-tailed noise | Heavy-tailed gradient noise | Yes |
| Zhang et al. (2024) | Hessian block heterogeneity | Block-diagonal Hessian structure | No |
| Tomihari & Sato (2025) | Gradient heterogeneity | Heterogeneous gradients and SignGD approx. | No |
| Xie et al. (2024) | $\ell_\infty$ geometry exploitation | Coordinate-wisely smooth w.r.t. $\ell_\infty$ norm | No |
| Kunstner et al. (2024) | Heavy-tailed class imbalance | Imbalanced data distribution | No |
| **Ours** | $v_t$-$g_t^2$ **decoupling** | **Loss landscape degeneracy** | No |

## 3.2. Algorithm Specifications

**Gradient Descent (GD) and Momentum.** Standard GD updates as $x_{t+1} = x_t - \eta \nabla L(x_t)$. Momentum adds a velocity term:

$$m_{t+1} = \beta m_t + \nabla L(x_t), \quad x_{t+1} = x_t - \eta m_{t+1},$$

where $\beta \in [0, 1)$ and $\eta$ is a constant learning rate.

**Adam.** Adam incorporates adaptive learning rates via exponential moving averages of gradient moments. Let $m_t$ and $v_t$ denote the first and second moment estimates:

$$m_t = \beta_1 m_{t-1} + (1 - \beta_1)g_t, \quad v_t = \beta_2 v_{t-1} + (1 - \beta_2)g_t^2,$$

where $g_t = \nabla L(x_t)$, and $\beta_1, \beta_2 \in [0, 1)$ are hyperparameters. The parameter update is:

$$x_{t+1} = x_t - \eta \frac{\hat{m}_t}{\sqrt{\hat{v}_t} + \varepsilon}, \quad (2)$$

where $\hat{m}_t = m_t/(1 - \beta_1^t)$ and $\hat{v}_t = v_t/(1 - \beta_2^t)$ are bias-correction terms, and $\varepsilon$ prevents division by zero.

**Special Cases of Adam.** (i) When $\beta_1 = 0$, Adam reduces to **RMSProp**: $x_{t+1} = x_t - \eta \cdot \frac{g_t}{\sqrt{v_t} + \varepsilon}$. (ii) When $\beta_1 = \beta_2 = 0$ and $\varepsilon \to 0$, Adam reduces to **SignGD**: $x_{t+1} = x_t - \eta \cdot \text{sgn}(g_t)$, which cannot converge with constant stepsize and stagnates at $O(L(\eta))$.

**Theoretical Simplifications.** To focus on asymptotic convergence and intrinsic dynamics, we adopt two standard simplifications:

- *Vanishing $\varepsilon$:* We take $\varepsilon = 0$ since non-zero $\varepsilon$ eventually dominates $\sqrt{v_t}$ as gradients vanish, reducing Adam to momentum-GD.

- *Asymptotic bias correction:* For large $t$, bias correction terms satisfy $1 - \beta^t \approx 1$, so we analyze uncorrected moments $m_t$ and $v_t$.

## 4. Adam Convergence on Highly Degenerate Polynomials

We derive the state space equations for Adam and present local convergence results on highly degenerate functions.

**State Space Dynamics of Adam.** For $L(x) = \frac{1}{k}x^k$, substituting $g_t = x_t^{k-1}$ into the Adam update rules with $\varepsilon = 0$ yields:

$$\begin{cases} m_t = \beta_1 m_{t-1} + (1 - \beta_1)x_t^{k-1}, \\ v_t = \beta_2 v_{t-1} + (1 - \beta_2)x_t^{2k-2}, \\ x_{t+1} = x_t - \eta \dfrac{m_t}{\sqrt{v_t}}. \end{cases}$$

To decouple the iterate scale from optimizer dynamics, we introduce normalized state variables:

$$\omega_t := \frac{m_t}{x_t^{k-1}}, \quad \lambda_t := \frac{x_t^{k-2}}{\sqrt{v_t}}, \quad (3)$$

where $\omega_t$ represents the normalized first moment and $\lambda_t \geq 0$ the *effective curvature*, capturing the relative scaling between Hessian-induced curvature and adaptive step size. The update rule becomes:

$$x_{t+1} = x_t - \eta \frac{m_t}{x_t^{k-1}} \cdot \frac{x_t^{k-1}}{\sqrt{v_t}} = (1 - \eta\omega_t\lambda_t)x_t. \quad (4)$$

The full dynamics are governed by:

$$\begin{cases} \omega_{t+1} = \dfrac{\beta_1 \omega_t}{(1 - \eta\omega_t\lambda_t)^{k-1}} + 1 - \beta_1 \\ \lambda_{t+1} = \dfrac{(1 - \eta\omega_t\lambda_t)^{k-2}\lambda_t}{\sqrt{\beta_2 + (1 - \beta_2)(1 - \eta\omega_t\lambda_t)^{2k-2}\lambda_t^2 x_t^2}} \\ x_{t+1} = (1 - \eta\omega_t\lambda_t)x_t \end{cases} \quad (5)$$

Loss decreases monotonically if and only if:

$$L(x_{t+1}) \leq L(x_t) \iff 0 \leq \omega_t\lambda_t \leq \frac{2}{\eta}. \quad (6)$$

**Why Adam Converges on Degenerate Functions.** Fig. 2 intuitively illustrates why Adam exhibits instability on strongly convex quadratics but converges naturally on degenerate landscapes. Fig. 2(a) shows that for $k = 2$, Adam fails with loss spikes (red shaded region), while for $k > 2$, it achieves stable exponential convergence. Fig. 2(b) reveals the mechanism through the effective curvature $\lambda_t$. For

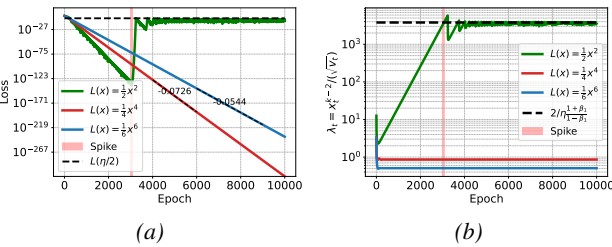

*Figure 2.* **(a)** Training loss of Adam ($\beta_1 = 0.9, \beta_2 = 0.93$) for different $L(x)$. The text online shows the slope of the exponential fit. **(b)** Evolution of effective curvature $\lambda_t$. The red shaded region highlights instability where $\lambda_t$ exceeds the theoretical stability threshold $\frac{2}{\eta} \frac{1+\beta_1}{1-\beta_1}$ (Cohen et al., 2023; Bai et al., 2025).

$k = 2$, $\lambda_t = 1/\sqrt{v_t}$ grows unbounded as $v_t$ decays, exceeding the stability threshold. However, for $k > 2$, the numerator $x_t^{k-2}$ and denominator $\sqrt{v_t}$ decay synchronously, causing $\lambda_t$ to converge to a constant, maintaining stability.

**Local Stability Analysis of Adam Convergence.** Motivated by the experiments in Fig. 2, we analyze the conditions for the convergence of the iterative system defined in Eq. (5). The system admits two classes of fixed points:

(i) The trivial fixed point: $(\omega^*, \lambda^*, x^*) = (1, 0, c)$ for any constant $c$, which is unstable.

(ii) The non-trivial fixed point, which corresponds to linear convergence:

$$\omega^* = \frac{1 - \beta_1}{1 - \beta_1 \beta_2^{-\frac{k-1}{2(k-2)}}}, \quad \lambda^* = \frac{1 - \beta_2^{\frac{1}{2(k-2)}}}{\eta \omega^*}, \quad x^* = 0. \tag{7}$$

Crucially, if the system converges to the non-trivial fixed point (ii), the product $\omega_t \lambda_t$ approaches the equilibrium value $\omega^* \lambda^* = \frac{1}{\eta}(1 - \beta_2^{\frac{1}{2(k-2)}})$. Since $\beta_2 < 1$, this value is strictly positive and remains well below the stability threshold $2/\eta$. Consequently, the system exhibits linear convergence. Within the basin of attraction, convergence hinges entirely on the *local asymptotic stability* of this fixed point.

This local stability is determined by the Jacobian matrix of the linearized system. The full Jacobian exhibits a block structure where the eigenvalue associated with the $x$-dynamics is explicitly $\beta_2^{\frac{1}{2(k-2)}} < 1$. Thus, stability is governed by the $2 \times 2$ sub-matrix $J$ describing the coupled $(\omega, \lambda)$ dynamics:

$$J = \begin{bmatrix} \beta_1 \Gamma(k-1)\beta_2^{\frac{-k}{2(k-2)}} + \beta_1 \beta_2^{\frac{1-k}{2(k-2)}} & \frac{\beta_1 \eta(1-\beta_1)^2 (k-1)\beta_2^{\frac{-k}{2(k-2)}}}{\left(1 - \beta_1 \beta_2^{-\frac{k-1}{2(k-2)}}\right)^2} \\ \frac{\Gamma^2(2-k)\left(1 - \beta_1 \beta_2^{-\frac{k-1}{2(k-2)}}\right)^2 \beta_2^{\frac{k-3}{2(k-2)}}}{\sqrt{\beta_2}\eta(1-\beta_1)^2} & \frac{\Gamma(2-k)\beta_2^{\frac{k-3}{2(k-2)}} + \sqrt{\beta_2}}{\sqrt{\beta_2}} \end{bmatrix}$$

where $\Gamma := 1 - \beta_2^{\frac{1}{2(k-2)}}$. We establish the following theorem to characterize this local stability:

**Theorem 4.1** (**Local Stability and Convergence Rate of Adam**). *Consider the dynamical system in Eq. (5) and hyperparameters $\beta_1, \beta_2 \in [0, 1)$. The following hold:*

(i) **Existence:** *The non-trivial fixed point* (7) *exists if and only if $\beta_1 < \beta_2^{\frac{k-1}{2(k-2)}}$.*

(ii) **Stability:** *The non-trivial fixed point is asymptotically stable, i.e., spectral radius $r(J) < 1$ if and only if:*

$$\beta_1 < \beta_2^{\frac{k}{2(k-2)}}, \quad \text{and} \tag{8}$$

$$\beta_1 > \frac{(k-2)\beta_2^{-\frac{1}{2(k-2)}} - k}{(k - (k-2)\beta_2^{\frac{1}{2(k-2)}})\beta_2^{-\frac{k}{2(k-2)}}}. \tag{9}$$

(iii) **Convergence Rate:** *Within the basin of attraction, iterates converge linearly:*

$$\lim_{t \to \infty} \frac{x_{t+1}}{x_t} = \beta_2^{\frac{1}{2(k-2)}} \tag{10}$$

*Proof.* The detailed derivation and rigorous proof are provided in Appendix A. $\square$

**Visualization of Theoretical and Empirical Phase Diagrams.** Fig. 3(a) visualizes the theoretical stability region ($r(J) < 1$) for $k = 4$, primarily governed by $\beta_1 < \beta_2^{\frac{k}{2(k-2)}}$ (simplifying to $\beta_1 < \beta_2$ for $k = 4$), with a minor constraint from the second inequality (lower-left corner). Fig. 3(b) shows strong empirical alignment: configurations satisfying the stability condition achieve machine precision (loss $\approx 10^{-300}$), while violations result in significantly higher losses. Furthermore, the theoretical loss convergence rate, derived as $\frac{k \ln \beta_2}{2(k-2)}$, yields values of $\approx -0.0726$ for $k = 4$ and $\approx -0.0544$ for $k = 6$. These predictions align precisely with the empirical slopes observed in Fig. 2(a).

*Remark* 4.2 (**Monotonic expansion of stability region.**). As degeneracy order $k$ increases, the exponent $\frac{k}{2(k-2)}$ in the primary stability constraint $\beta_1 < \beta_2^{\frac{k}{2(k-2)}}$ decreases. Since $\beta_2 < 1$, this relaxes the upper bound on $\beta_1$, enlarging the stable region. Thus, satisfying the condition for $k = 4$ guarantees stability for all $k > 4$.

*Remark* 4.3 (**Basin of Attraction for Full Adam.**). Characterizing the exact global basin of attraction for Thm. 4.1 is challenging due to the strong nonlinear coupling between $\omega_t$ and $\lambda_t$ introduced by the momentum $m_t$. This coupling transforms the process into a 3-dimensional discrete dynamical system over $(x_0, m_0, v_0)$ with a potentially complex, initialization-dependent boundary, unlike the simpler, monotonic state space of RMSProp (see Thm. 5.7). Nevertheless, empirical evidence demonstrates a broad and robust

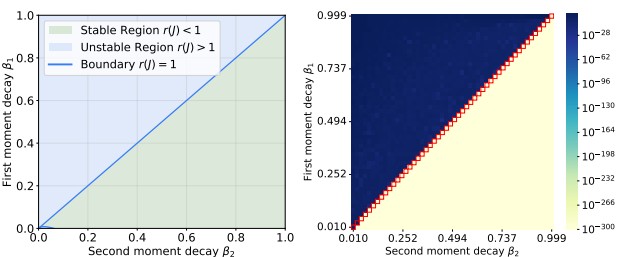

*(a)* Theoretical Phase Diagram    *(b)* Empirical Phase Diagram

*Figure 3.* **(a)** Theoretical convergence phase diagram for $k = 4$, partitioned according to the stability conditions derived in Eq. (8) and Eq. (9). **(b)** Experimental validation using Adam on $L(x) = \frac{1}{4}x^4$ with $x_0 = 1.0, \eta = 0.001$. The heatmap displays the final training loss after $100,000$ steps. Additional results for $k = 6$ can be found in Fig. 11. For a detailed discussion and a magnified view of the lower-left corner region, please refer to Appendix G.1.

effective basin. As shown in Fig. 3(b), under standard initializations ($m_0 = g_0$, $v_0 = g_0^2$), the convergence region aligns closely with our local stability predictions, allowing typical trajectories to reliably enter the theoretical linear convergence basin.

*Remark* 4.4 (**Extension to General Degenerate Minima**). While Thm. 4.1 is explicitly derived for the monomial loss $L(x) = \frac{1}{k}x^k$, the theoretical results extend directly to a broader class of functions $L(x) = \frac{1}{k}x^k(1 + h(x))$ for $k \geq 4$, where $h$ is an analytic function satisfying $h(0) = 0$. For this generalized function class, the gradient is given by $g(x) = x^{k-1}\left(1 + h(x) + \frac{1}{k}xh'(x)\right) = \Theta(x^{k-1})$ as $x \to 0$. Because the higher-order terms introduced by $h$ vanish when evaluating the Jacobian at the fixed point $(\omega^*, \lambda^*, 0)$, the local stability conditions (Eqs. (8), (9)) and the resulting convergence rate (Eq. (10)) remain perfectly identical. Consequently, our theoretical framework captures the universal local behavior of the optimizer around any $k$-th order degenerate minimum, rather than being restricted to purely monomial landscapes.

# 5. The Power of Adaptivity: Exponential Speedup Against Degeneration

Degeneracy in objective functions often degrades convergence rates from linear to sublinear. We demonstrate that while GD and momentum are confined to slow sublinear convergence on degenerate landscapes, adaptive mechanisms achieve linear convergence acceleration through implicit amplification of the effective learning rate.

## 5.1. The Curse of Degeneracy: Slow Convergence of GD and Momentum

We establish that both GD and Momentum suffer from *power-law* convergence on degenerate objectives.

**Theorem 5.1** (**Power-Law Convergence of Gradient Flow**). *Consider gradient flow on $L(x) = \frac{1}{k}x^k$ ($k \geq 4$, even): $\dot{x} = -\eta\nabla L(x) = -\eta x^{k-1}$, $x(0) = x_0 \neq 0$. For any fixed learning rate $\eta > 0$, the solution $x(t)$ exhibits an asymptotic power-law decay:*

$$x(t) = \Theta\left(t^{-\frac{1}{k-2}}\right) \quad as \ t \to \infty. \tag{11}$$

*See Appendix C.1 for the proof.*

*Remark* 5.2 (**The Curse of Degeneracy**). Theorem 5.1 reveals a severe "*curse of degeneracy*" regarding iteration complexity. Inverting the rate $x(t) \sim t^{-\frac{1}{k-2}}$ shows that reaching precision $|x(t)| \leq \epsilon$ requires time: $T_\epsilon \sim \epsilon^{-(k-2)} = \left(\frac{1}{\epsilon}\right)^{k-2}$, implying computational cost grows **exponentially** with degeneracy order $k$.

We show that momentum provides no asymptotic improvement in degenerated settings.

**Theorem 5.3** (**Power-Law Convergence of Momentum**). *Consider continuous-time Heavy-ball momentum with $\beta_1 \in [0, 1)$: $\dot{m} = -(1 - \beta_1)m + (1 - \beta_1)\nabla L(x)$, $m(0) = 0$, and $\dot{x} = -\eta m$. For $L(x) = \frac{1}{k}x^k$ ($k \geq 4$, even), the trajectory $x(t)$ satisfies:*

$$x(t) = \Theta\left(t^{-\frac{1}{k-2}}\right) \quad as \ t \to \infty. \tag{12}$$

*See Appendix C.2 for the proof.*

Thm. 5.3 confirms momentum fails to break the curse of degeneracy. As curvature vanishes ($L''(x) \propto x^{k-2} \to 0$), the gradient force diminishes too rapidly to sustain momentum, resulting in the same complexity class as GD: $T_\epsilon \sim \epsilon^{-(k-2)}$.

## 5.2. Exponentially Amplified Learning Rate Mechanism

To isolate adaptivity effects, we analyze RMSProp (Adam with $\beta_1 = 0$) and demonstrate that adaptive rescaling mitigates vanishing curvature, thereby accelerating convergence from sublinear to linear.

Consider RMSProp with $\varepsilon = 0$:

$$v_t = \beta_2 v_{t-1} + (1 - \beta_2)g_t^2, \tag{13}$$

$$x_{t+1} = x_t - \eta\frac{g_t}{\sqrt{v_t}}. \tag{14}$$

The core intuition is that as $x_t$ converges, the update term $\frac{g_t}{\sqrt{v_t}}$ must vanish, causing the second moment update Eq. (13) to be dominated by the former memory term. This induces a geometric decay in $v_t$, which in turn drives an exponential increase in the *effective learning rate* $\eta_{\text{eff},t} := \eta/\sqrt{v_t}$.

**Lemma 5.4** (**Asymptotic Behavior of Second Moments**). *For RMSProp dynamics in Eq. (14), if $\{x_t\}$ converges, then:*

$$\lim_{t \to \infty} \frac{v_t}{v_{t-1}} = \beta_2. \tag{15}$$

*Proof.* Convergence implies $\lim_{t\to\infty} \frac{g_t}{\sqrt{v_t}} = 0$, hence $\lim_{t\to\infty} \frac{g_t^2}{v_t} = 0$. Rearranging (13) as $g_t^2 = \frac{v_t - \beta_2 v_{t-1}}{1 - \beta_2}$. $\lim_{t\to\infty} \frac{v_t - \beta_2 v_{t-1}}{(1-\beta_2)v_t} = 0$ implies $\lim_{t\to\infty} \left(1 - \beta_2 \frac{v_{t-1}}{v_t}\right) = 0$. Thus, $\lim_{t\to\infty} \frac{v_t}{v_{t-1}} = \beta_2$. $\square$

Lem. 5.4 reveals the acceleration mechanism: $v_t \sim \beta_2^t$ implies $\eta_{\text{eff},t} \propto \beta_2^{-t/2}$. This exponential schedule accelerates convergence from polynomial to exponential.

**Lemma 5.5** (**Linear Convergence via Exponential Learning Rate**). *Consider gradient flow on $L(x) = \frac{1}{k}x^k$ ($k \geq 4$, even) with time-varying learning rate $\dot{x} = -\eta(t)x^{k-1}$. If the learning rate follows an exponential schedule $\eta(t) = \eta_0 e^{\alpha t}$ ($\alpha > 0$), the solution $x(t)$ exhibits asymptotic exponential convergence:*

$$x(t) \sim C \exp\left(-\frac{\alpha}{k-2}t\right) \quad as\ t \to \infty, \qquad (16)$$

*where $C$ is a constant determined by system parameters. See Appendix C.3 for the proof.*

### 5.3. Discretization Introduces Stability Constraints

While Lem. 5.5 suggests exponential schedules yield unbounded acceleration, discretization imposes stability constraints. The learning rate growth factor $\gamma := e^\alpha$ is upper-bounded by local curvature dynamics.

Consider gradient descent on $L(x) = \frac{1}{k}x^k$ with exponentially increasing learning rate:

$$\begin{aligned} x_{t+1} &= x_t - \eta_t x_t^{k-1} = (1 - \eta_t x_t^{k-2})x_t, \\ \eta_{t+1} &= \gamma \eta_t, \quad (\gamma > 1). \end{aligned} \qquad (17)$$

We introduce the *effective sharpness* $u_t := \eta_t x_t^{k-2}$, capturing the interplay between step size and local Hessian. If $u_t > 2$, the step size exceeds stability limits of the local curvature, causing instability. Substituting the update rules Eq. (17) yields a closed-form recurrence relation for $u_t$, reducing the two-dimensional dynamics to a one-dimensional map:

$$u_{t+1} = \psi(u_t) := \gamma u_t (1 - u_t)^{k-2}. \qquad (18)$$

Fig. 4 shows the limit set of $u_t$ for varying $\gamma$ after a 500-iteration transient. For small $\gamma$, the system converges to a stable fixed point. As $\gamma$ increases, period-doubling bifurcations occur, eventually transitioning to chaos; however, $x_t$ continues to converge because $u_t < 2$. With further increases in $\gamma$, the trajectory of $u_t$ has sharp transition escaping the stable region $(0, 2)$ (i.e., diverges) after 500 iterations.

We formalize the stability condition corresponding linear convergence with a fixed rate in the following proposition.

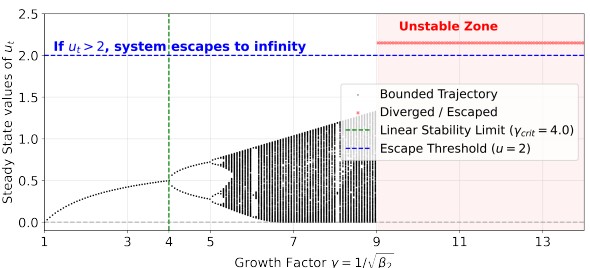

*Figure 4.* **Bifurcation diagram of effective sharpness $u_t$ for $k = 4$.** The system Eq: (18) exhibits three regimes: (I) **Stable Fixed Point** ($u_t \to u^*$), leading to exponential convergence of $x_t$ with a fixed rate; (II) **Period-Doubling** ($u_t$ oscillates within $(0, 2)$), preserving convergence albeit with oscillating rates; (III) **Divergence** ($u_t > 2$), where numerical instability occurs.

**Proposition 5.6** (**Stability of the Acceleration Map**). *The map in Eq. (18) has a non-zero fixed point $u^* = 1 - \gamma^{-\frac{1}{k-2}} \in (0,1)$, locally asymptotically stable if and only if:*

$$\gamma < \gamma_{crit} := \left(\frac{k}{k-2}\right)^{k-2}. \qquad (19)$$

*See Appendix D for Proof.*

For RMSProp, $\gamma = 1/\sqrt{\beta_2}$, so stability requires:

$$\beta_2 > \beta_2^{\text{crit}} := \left(\frac{k-2}{k}\right)^{2(k-2)}. \qquad (20)$$

For $k = 4$, this gives $\beta_2 > 0.0625$. Since practical values are typically $\beta_2 \in [0.9, 0.999]$, standard adaptive optimizers operate within the stable regime. Under this stability condition, we establish the global convergence of RMSProp.

**Theorem 5.7** (**Global Linear Convergence of RMSProp**). *Consider minimizing $L(x) = \frac{1}{k}x^k$ ($k \geq 4$, even) using RMSProp. Assume the hyperparameter satisfies the stability condition $\beta_2 > (\frac{k-2}{k})^{2(k-2)}$ and the initialization satisfies $u_0 = \eta x_0^{k-2}/\sqrt{v_0} < 1$. Then:*

*(i) **Convergence:** The iterates $\{x_t\}$ converge to $x^* = 0$.*

*(ii) **Linear Rate:** Convergence is linear:*

$$\lim_{t\to\infty} \frac{x_{t+1}}{x_t} = \beta_2^{\frac{1}{2(k-2)}} \qquad (21)$$

*See Appendix B for Proof.*

*Remark* 5.8 (**Fundamental Complexity Separation**). Thm. 5.7 reveals a fundamental separation: GD requires iteration complexity $T_\epsilon \sim \epsilon^{-(k-2)}$ (exponential, exploding with $k$), while adaptive methods achieves **linear complexity** $T_\epsilon \sim (k-2)\ln(1/\epsilon)$. Adaptivity flattens the complexity class, reducing dependence on $k$ from exponential to linear.

# 6. Phase Diagram of Adam Training Behaviors

Adam's non-convergence has been empirically observed to exhibit oscillations or significant spikes (Ma et al., 2022). We classify behavior under different hyperparameters by examining the fixed-point properties. Following Sec. 5.3, we present the phase diagram characterizing Adam's fixed-point properties under $\beta_2 > \left(\frac{k-2}{k}\right)^{2(k-2)}$.

**Theorem 6.1** (**Phase Transitions with Fixed-Point Behavior**). *Consider the three-dimensional dynamical system governing Adam in Eq.* (5). *Let* $\beta_2 > \left(\frac{k-2}{k}\right)^{2(k-2)}$. *The following properties hold:*

(i) *If* $\beta_1 > \beta_2^{\frac{k-1}{2(k-2)}}$, *no non-trivial fixed point exists; the only fixed points are the trivial solutions* $(\omega^*, \lambda^*, x^*) = (1, 0, c)$ *for any constant* $c$.

(ii) *If* $\beta_2^{\frac{k}{2(k-2)}} < \beta_1 < \beta_2^{\frac{k-1}{2(k-2)}}$, *the non-trivial fixed point* (7) *exists but is asymptotically unstable.*

(iii) *If* $\beta_1 < \beta_2^{\frac{k}{2(k-2)}}$, *the non-trivial fixed point* (7) *exists and is asymptotically stable.*
*See Appendix A.2 for Proof.*

Fig. 5 illustrates the phase diagram with three distinct regions. Notably, although the non-trivial fixed point is unstable in certain regimes, its existence can still attract parameters effectively during the early phase. This leads to a transient convergence followed by an escape, manifesting as the "spike" phenomenon discussed in Sec. 6.1.

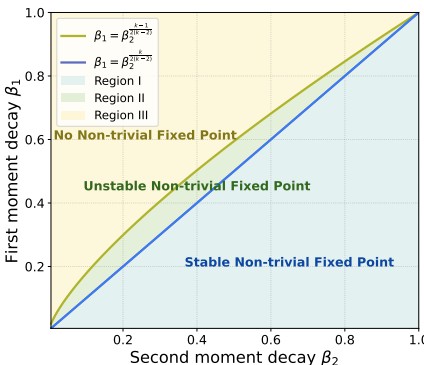

*Figure 5.* Phase diagram for $k = 4$ according to Thm. 6.1.

## 6.1. Typical Cases over the Phase Diagram

Fig. 6 illustrates typical behaviors for $\beta_1 = 0.9$ with $\beta_2 \in \{0.91, 0.895, 0.8\}$ (see Fig. 14 for additional examples). While $m_t$ and $g_t$ remain tightly coupled across all cases, $v_t$ and $g_t^2$ exhibit distinct coupling patterns.

**Regime I: Stable Exponential Convergence.** Occurs when $\beta_1 < \beta_2^{\frac{k}{2(k-2)}}$ (stable non-trivial fixed point, Fig. 6(a, d, g)).

- **Behavior:** The loss converges exponentially to machine precision.
- **Mechanism (Decoupling):** As $g_t$ rapidly vanishes, $v_t$ decouples from $g_t^2$ and follows autonomous decay: $v_t \approx \beta_2 v_{t-1}$. Fig. 6 (d) shows $\log v_t$ slope matching $\ln(\beta_2) \approx -0.0943$. By Thm. 5.7, this geometric decay acts as an exponential learning rate schedule, predicting loss convergence rate $\frac{k \ln \beta_2}{2(k-2)} \approx -0.0943$, precisely matching Fig. 6 (a).

**Regime II: Exponential Convergence with Late-Phase Spikes.** Occurs when $\beta_2^{\frac{k-1}{2(k-2)}} < \beta_1 < \beta_2^{\frac{k}{2(k-2)}}$ (unstable non-trivial fixed point, Fig. 6 (b, e, h)).

- **Behavior:** Initial exponential convergence below the SignGD threshold (dashed line $L(\eta/2)$), interrupted by a violent loss spike.
- **Mechanism (Decoupling and Instability):** Like Regime I, $v_t$ initially decouples from $g_t^2$ (Fig. 6(e) shows decay slope $\approx \ln \beta_2$), driving acceleration. Fig. 6 (h) confirms that the parameter trajectory initially evolves around the nontrivial fixed point (see also Fig. 14(h); intuitive explanation in Appendix E). However, fixed-point instability triggers $\omega_t \lambda_t > 2/\eta$ (Fig. 6(e) subplot), causing $|x_t|$ divergence. Since $\sqrt{v_t}$ continues autonomous decay, the scaling $\lambda_t := \frac{x_t^{k-2}}{\sqrt{v_t}}$ grows rapidly. The pathology arises from **response delay**: $v_t$ requires several iterations to re-couple with the surging $g_t^2$, manifesting as a transient spike, consistent with the findings of Bai et al. (2025)

**Regime III: SignGD-like Oscillation.** Occurs when $\beta_1 > \beta_2^{\frac{k-1}{2(k-2)}}$ (no non-trivial fixed point, Fig. 6 (c, f, i)).

- **Behavior:** No exponential convergence; loss saturates around $L(\eta/2)$ with oscillatory behavior.
- **Mechanism (Tight Coupling):** For small $\beta_2$, fast $v_t$ adaptation keeps it coupled with $g_t^2$ (Fig. 6(f)). Since $v_t \sim g_t^2$, the adaptive step $\eta/\sqrt{v_t} \sim \eta/|g_t|$ cancels gradient magnitude, yielding $x_{t+1} \approx x_t - \eta \cdot \text{sgn}(g_t)$. This provides **instantaneous negative feedback**: when instability triggers ($\omega_t \lambda_t > 2/\eta$, Fig. 6(f) subplot) and $|g_t|$ increases, $v_t$ rises simultaneously, suppressing instability into low-amplitude oscillations.

## 6.2. Experimental Demonstration of the Phase Diagram

We verify theoretical phase diagram through grid search over $\beta_1, \beta_2$ on $L(x) = \frac{1}{4}x^4$, recording minimum loss $\min_t L(x_t)$ and final loss $L(x_T)$. Fig. 7(a-b) shows three regions (yellow = low loss, blue = high loss) aligning with theoretical predictions:

- **Regime I (Stable):** When $\beta_1 < \beta_2^{1.0}$ (for $k = 4$), both minimum and final losses are low (yellow), confirming

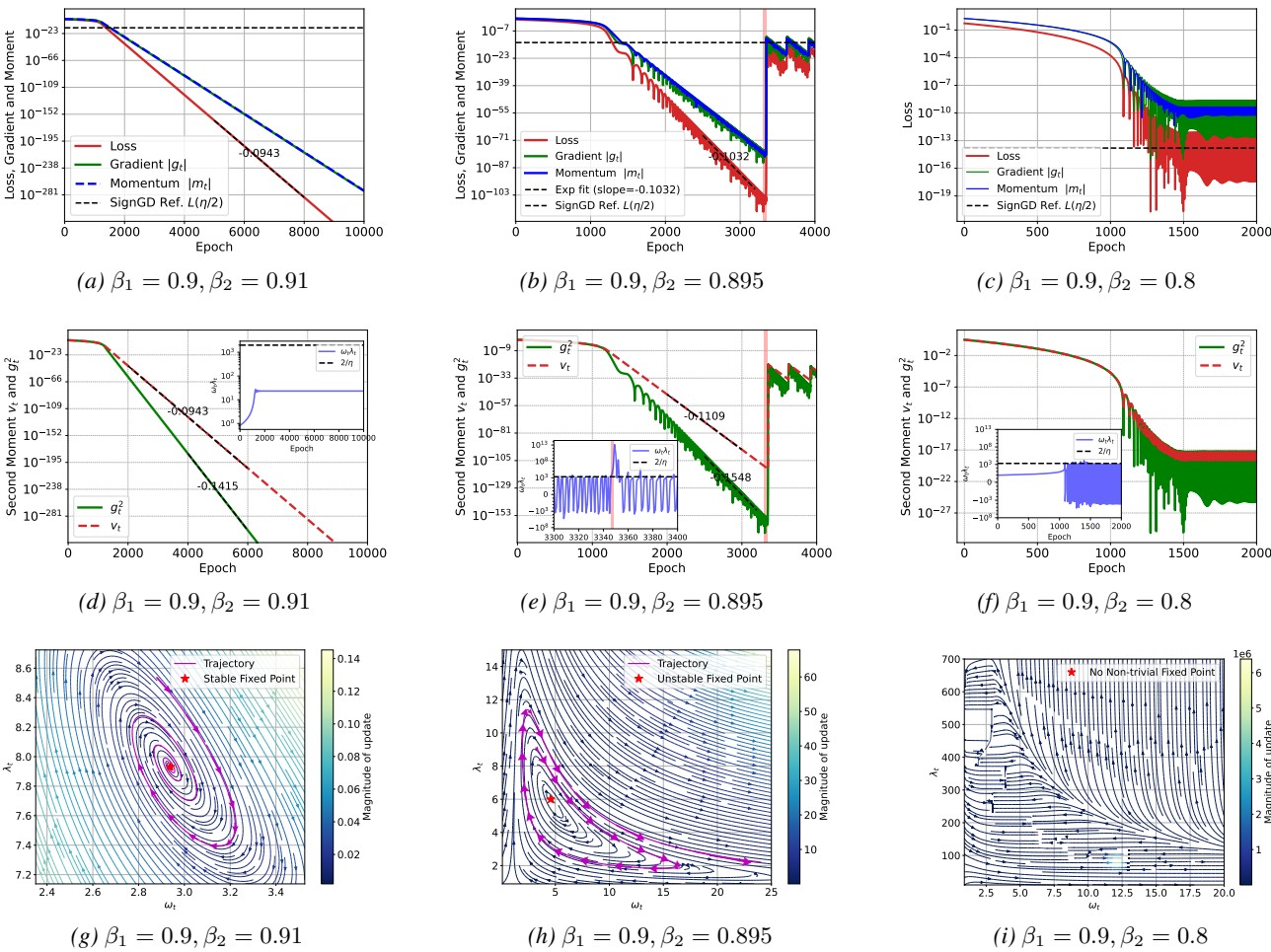

*Figure 6.* Three dynamical behaviors of Adam on $L(x) = \frac{1}{4}x^4$ with $\eta = 0.001$. **Top Row:** Evolution of training loss, gradient and first moment. The text online shows the slope of the exponential fit. **Middle Row:** Evolution of the second moment estimate $v_t$ versus the squared gradient $g_t^2$. Insets show the evolution of the stability metric $\omega_t\lambda_t$ relative to the theoretical threshold $2/\eta$. **Bottom Row:** Vector fields of $\omega_t$ and $\lambda_t$; purple lines represent evolutionary trajectories. **(Left) Regime I: Stable Exponential Convergence.** $v_t$ decouples from $g_t^2$, enabling stable exponential acceleration. **(Middle) Regime II: Exponential Convergence with Spike.** $v_t$ initially decouples, driving acceleration toward the fixed point. However, drastic violation of the stability condition ($\omega_t\lambda_t > 2/\eta$) due to fixed-point instability triggers a loss spike. **(Right) Regime III: SignGD-like Oscillation.** $v_t$ tracks $g_t^2$ closely (tight coupling), preventing exponential convergence. Violations of the stability threshold trigger instantaneous corrections, resulting in oscillations.

stable exponential convergence.

- **Regime II (Spikes):** For $\beta_2^{1.0} < \beta_1 < \beta_2^{0.75}$, Fig. 7(a) shows low minimum loss (transient convergence) while Fig. 7(b) shows high final loss (spike mechanism).

- **Regime III (Oscillation):** When $\beta_1 > \beta_2^{0.75}$, both losses remain high, indicating no exponential phase.

To definitively attribute the acceleration in Regimes I and II to the decoupling mechanism, we also visualize the coupling ratio $R_t^{(v)} = v_t/g_t^2$. Fig. 7(c-d) shows coupling ratio $R_t^{(v)}$. In Regimes I and II, $\max_t R_t^{(v)}$ reaches high values, confirming decoupling drives acceleration and low loss. In Regime III, the ratio stays low ($\approx 1$, blue), confirming tight coupling prevents exponential speedup.

## 7. Discussion and Conclusion

**Coupling of Degenerate and Non-Degenerate Modes.** High-dimensional optimization landscapes often couple quadratic and higher-order (degenerate) terms. Fig. 8 demonstrates Adam's distinct advantages over GD and Momentum in such mixed-curvature scenarios. When degenerate directions are present, GD and Momentum are bottlenecked by slow, sublinear convergence, whereas Adam maintains rapid optimization. However, quadratic components can induce instability, manifesting as loss spikes (see the detailed analysis of quadratic case in Appendix F). These spikes, while disrupting monotonicity, stem from the same aggressive scaling that enables Adam to escape degenerate

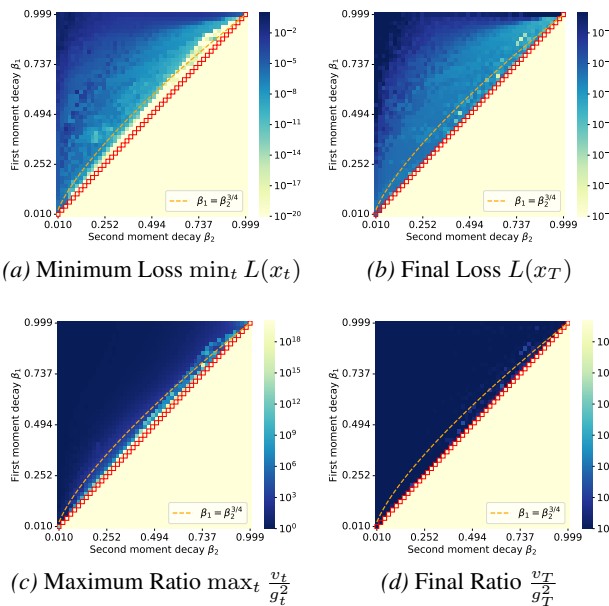

*(a)* Minimum Loss $\min_t L(x_t)$

*(b)* Final Loss $L(x_T)$

*(c)* Maximum Ratio $\max_t \frac{v_t}{g_t^2}$

*(d)* Final Ratio $\frac{v_T}{g_T^2}$

*Figure 7.* **Empirical phase diagram on** $L(x) = \frac{1}{4}x^4$. Sliding average filter (window size 10) applied for smoothing. **(a)** Minimum loss during training. **(b)** Final loss shows Regime II instability (blue sub-region of (a)). **(c-d)** Coupling ratios confirm exponential regions coincide with $v_t$-$g_t^2$ decoupling.

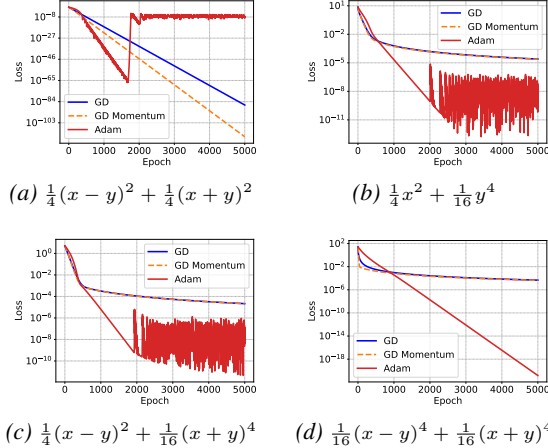

*(a)* $\frac{1}{4}(x-y)^2 + \frac{1}{4}(x+y)^2$

*(b)* $\frac{1}{4}x^2 + \frac{1}{16}y^4$

*(c)* $\frac{1}{4}(x-y)^2 + \frac{1}{16}(x+y)^4$

*(d)* $\frac{1}{16}(x-y)^4 + \frac{1}{16}(x+y)^4$

*Figure 8.* Optimization trajectories under different couplings of quadratic and quartic loss functions.

plateaus efficiently. In practice, learning rate decay readily mitigates this instability in quadratic directions while preserving Adam's speed advantage in degenerate regions.

**Relevance to Deep Learning Optimization.** Adam's advantages over GD vary significantly across architectures: more pronounced in Transformers for language tasks, less so in CNNs for vision tasks (Zhang et al., 2024). Our work suggests this may relate to differences in landscape degeneracy. Our preliminary experiments show that replacing

ReLU with softmax in MLPs significantly slows well-tuned GD compared to Adam (Fig. 15(a-b)), with softmax increasing degeneracy as evidenced by more concentrated spectral density at the origin (Fig. 15(c)). Furthermore, Fig. 15(d-f) shows Transformers exhibit higher degeneracy than CNNs on real data, correlating with Adam's larger advantage. This provides valuable directions for future investigation into architecture-dependent optimizer performance.

**Conclusion.** In this work, we investigate Adam's convergence on highly degenerate polynomials, identifying a class where Adam converges automatically without decaying learning rate schedulers. We prove Adam achieves local linear convergence on these degenerated functions, significantly outperforming the sublinear rates of GD and Momentum. We characterize the decoupling mechanism between $v_t$ and $g_t^2$ that exponentially amplifies the effective learning rate, and map Adam's hyperparameter phase diagram across three distinct regimes: stable convergence, spikes, and SignGD-like oscillation. Given the prevalence of degenerate directions in deep learning landscapes, our findings provide valuable insights into Adam's advantages. Future work includes investigating stochastic batch settings and connections to high-dimensional coupled optimization in real-world scenarios.

# Acknowledgements

This work was supported by the National Key R&D Program of China (Grant No. 2022YFA1008200), the National Natural Science Foundation of China (Grant Nos. 92270001, 12371511, 12422119, and 12571567), the Natural Science Foundation of Shanghai (Grant No. 25ZR1402280), and the 2025 Key Technology R&D Program "New Generation Information Technology" Project of the Shanghai Municipal Science and Technology Commission (Grant No. 25511103100).

# Impact Statement

This paper presents work whose goal is to advance the field of Machine Learning. There are many potential societal consequences of our work, none which we feel must be specifically highlighted here.

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

# A. Proof of Adam Local Stability Theorem 4.1

## A.1. Prerequisites: Local Stability Theory

In this section, we briefly review the local stability theory for dynamical systems. The analysis of local stability is primarily based on the linearization of the system near its equilibrium states, utilizing the Jacobian matrix.

Let $\Omega \subseteq \mathbb{R}^n$ be the state space and $\boldsymbol{x} \in \Omega$ be the state vector. We denote the Jacobian matrix of a vector-valued function $\boldsymbol{f} : \mathbb{R}^n \to \mathbb{R}^n$ evaluated at a point $\boldsymbol{x}^*$ as $J(\boldsymbol{x}^*) = D\boldsymbol{f}(\boldsymbol{x}^*)$. The eigenvalues of $J(\boldsymbol{x}^*)$ are denoted by $\lambda_i$ ($i = 1, \ldots, n$).

### A.1.1. DISCRETE-TIME SYSTEMS

Consider a discrete dynamical system governed by the difference equation:

$$\boldsymbol{x}_{k+1} = \boldsymbol{f}(\boldsymbol{x}_k). \tag{22}$$

A point $\boldsymbol{x}^*$ is called a *fixed point* if it satisfies $\boldsymbol{f}(\boldsymbol{x}^*) = \boldsymbol{x}^*$. According to linear stability theory, the fixed point $\boldsymbol{x}^*$ is locally asymptotically stable if all eigenvalues of the Jacobian matrix lie inside the unit circle in the complex plane. That is, the spectral radius $\rho(J)$ must satisfy:

$$\rho(J(\boldsymbol{x}^*)) = \max_{1 \leq i \leq n} |\lambda_i| < 1. \tag{23}$$

### A.1.2. CONTINUOUS-TIME SYSTEMS

Consider a continuous dynamical system described by the ordinary differential equation:

$$\dot{\boldsymbol{x}}(t) = \boldsymbol{f}(\boldsymbol{x}(t)). \tag{24}$$

A point $\boldsymbol{x}^*$ is an *equilibrium point* if $\boldsymbol{f}(\boldsymbol{x}^*) = \boldsymbol{0}$. The equilibrium $\boldsymbol{x}^*$ is locally asymptotically stable if all eigenvalues of the Jacobian matrix have strictly negative real parts:

$$\operatorname{Re}(\lambda_i) < 0, \quad \forall i = 1, \ldots, n. \tag{25}$$

Essentially, the stability criteria require the eigenvalues to be bounded by the unit circle for discrete maps, and by the imaginary axis for continuous flows.

## A.2. Local Stability Analysis

**Definition A.1** (Adam)**.**

$$m_t = \beta_1 m_{t-1} + (1 - \beta_1)g_t$$
$$v_t = \beta_2 v_{t-1} + (1 - \beta_2)g_t^2$$
$$x_{t+1} = x_t - \eta \cdot \frac{m_t}{\sqrt{v_t}}$$

Here, $g_t = \nabla L(x_t)$, where $L(x)$ is the loss function. $\beta_1, \beta_2$ are hyperparameters in $[0, 1)$. $\eta > 0$ is the learning rate.

**Theorem A.2.** *Consider using Adam in Definition A.1 to learn the loss function $L(x) = \frac{1}{k}x^k$, where $k$ is an even number no less than four. Change variables by defining $\omega_t = \frac{m_t}{g_t} = \frac{m_t}{x_t^{k-1}}, \lambda_t = \frac{x_t^{k-2}}{\sqrt{v_t}}$. Then the following statements hold:*

*1. The new variables obey the following dynamics:*

$$\omega_{t+1} = \frac{\beta_1 \omega_t}{(1 - \eta \omega_t \lambda_t)^{k-1}} + 1 - \beta_1$$

$$\lambda_{t+1} = \frac{(1 - \eta \omega_t \lambda_t)^{k-2} \lambda_t}{\sqrt{\beta_2 + (1 - \beta_2)(1 - \eta \omega_t \lambda_t)^{2k-2} \lambda_t^2 x_t^2}} \tag{26}$$

$$x_{t+1} = (1 - \eta \omega_t \lambda_t) x_t$$

*2. Dynamics (26) has two fixed points:*

$$\omega^* = 1, \lambda^* = 0, x^* \in \mathbb{R} \quad \text{and}$$

$$\omega^* = \frac{1 - \beta_1}{1 - \beta_1 \beta_2^{-\frac{k-1}{2(k-2)}}}, \lambda^* = \frac{1 - \beta_2^{\frac{1}{2(k-2)}}}{\eta \omega^*}, x^* = 0.$$

3. The fixed point $\omega^* = \dfrac{1-\beta_1}{1-\beta_1\beta_2^{-\frac{k-1}{2(k-2)}}}, \lambda^* = \dfrac{1-\beta_2^{\frac{1}{2(k-2)}}}{\eta\omega^*}, x^* = 0$ is meaningful (i.e., $\lambda^* \geq 0$) if and only if $\beta_1 < \beta_2^{\frac{k-1}{2(k-2)}}$.

4. The fix point $\omega^* = 1, \lambda^* = 0, x^* = s$ is unstable for all $s \in \mathbb{R}$.

5. Define the matrix $J$ to be

$$
\begin{bmatrix}
\beta_1\left(1-\beta_2^{\frac{1}{2(k-2)}}\right)(k-1)\beta_2^{\frac{-k}{2(k-2)}} + \beta_1\beta_2^{\frac{1-k}{2(k-2)}} & \dfrac{\beta_1\eta(1-\beta_1)^2(k-1)\beta_2^{\frac{-k}{2(k-2)}}}{\left(-\beta_1\beta_2^{-\frac{k-1}{2(k-2)}}+1\right)^2} \\[4ex]
\dfrac{\left(1-\beta_2^{\frac{1}{2(k-2)}}\right)^2(2-k)\left(-\beta_1\beta_2^{-\frac{k-1}{2(k-2)}}+1\right)^2\beta_2^{\frac{k-3}{2(k-2)}}}{\sqrt{\beta_2}\eta(1-\beta_1)^2} & \dfrac{\left(1-\beta_2^{\frac{1}{2(k-2)}}\right)(2-k)\beta_2^{\frac{k-3}{2(k-2)}}+\sqrt{\beta_2}}{\sqrt{\beta_2}}
\end{bmatrix}
$$

Then the matrix $\begin{bmatrix} J & 0 \\ 0 & \beta_2^{\frac{1}{2(k-2)}} \end{bmatrix}$ is the Jacobian matrix at the fixed point $\omega^* = \dfrac{1-\beta_1}{1-\beta_1\beta_2^{-\frac{k-1}{2(k-2)}}}, \lambda^* = \dfrac{1-\beta_2^{\frac{1}{2(k-2)}}}{\eta\omega^*}, x^* = 0$.
Let $r(J)$ be the spectral radius of $J$.
- If $r(J) < 1$, then the fixed point is asymptotically stable.
- If $r(J) > 1$, then the fixed point is unstable.

6. $r(J) < 1$ if and only if both of the two inequalities hold:

$$\beta_1 < \beta_2^{\frac{k}{2(k-2)}}$$

$$\beta_1 > \dfrac{(k-2)\beta_2^{-\frac{1}{2(k-2)}} - k}{(k-(k-2)\beta_2^{\frac{1}{2(k-2)}})\beta_2^{-\frac{k}{2(k-2)}}}.$$

Besides, $r(J) = 1$ if and only if at least one of these two inequalities becomes an equality. As a consequence, when $\beta_2 > (\frac{k-2}{k})^{2(k-2)}$, $r(J) < 1$ if and only if $\beta_1 < \beta_2^{\frac{k}{2(k-2)}}$.

7. When $\beta_1 < \beta_2^{\frac{k}{2(k-2)}}$ and $\beta_1 > \dfrac{(k-2)\beta_2^{-\frac{1}{2(k-2)}}-k}{(k-(k-2)\beta_2^{\frac{1}{2(k-2)}})\beta_2^{-\frac{k}{2(k-2)}}}$, the fixed point $\omega^* = \dfrac{1-\beta_1}{1-\beta_1\beta_2^{-\frac{k-1}{2(k-2)}}}, \lambda^* = \dfrac{1-\beta_2^{\frac{1}{2(k-2)}}}{\eta\omega^*}, x^* = 0$ is asymptotically stable, and $\lim_{t\to\infty}\dfrac{x_{t+1}}{x_t} = \beta_2^{\frac{1}{2(k-2)}}$.

8. When $\beta_1 > \beta_2^{\frac{k}{2(k-2)}}$ or $\beta_1 < \dfrac{(k-2)\beta_2^{-\frac{1}{2(k-2)}}-k}{(k-(k-2)\beta_2^{\frac{1}{2(k-2)}})\beta_2^{-\frac{k}{2(k-2)}}}$, the fixed point $\omega^* = \dfrac{1-\beta_1}{1-\beta_1\beta_2^{-\frac{k-1}{2(k-2)}}}, \lambda^* = \dfrac{1-\beta_2^{\frac{1}{2(k-2)}}}{\eta\omega^*}, x^* = 0$ is unstable,

*Proof.* **Proof of Statements** 1-4:

Consider Adam:

$$m_t = \beta_1 m_{t-1} + (1-\beta_1)x_t^{k-1}$$
$$v_t = \beta_2 v_{t-1} + (1-\beta_2)x_t^{2k-2}$$
$$x_{t+1} = x_t - \eta \cdot \frac{m_t}{\sqrt{v_t}}$$

Then we have:

$$x_{t+1} = (1 - \eta\frac{m_t}{x_t\sqrt{v_t}})x_t = (1 - \eta\frac{m_t}{x_t^{k-1}} \times \frac{x_t^{k-2}}{\sqrt{v_t}})x_t$$

Define

$$\omega_t = \frac{m_t}{x_t^{k-1}}, \quad \lambda_t = \frac{x_t^{k-2}}{\sqrt{v_t}}.$$

Then it holds that:

$$x_{t+1} = (1 - \eta\omega_t\lambda_t)x_t$$

$$\omega_{t+1} = \frac{m_{t+1}}{x_{t+1}^{k-1}} = \frac{\beta_1 m_t + (1-\beta_1)x_{t+1}^{k-1}}{x_{t+1}^{k-1}} = \frac{\beta_1 m_t}{x_{t+1}^{k-1}} + 1 - \beta_1 = \frac{\beta_1\omega_t}{(1-\eta\omega_t\lambda_t)^{k-1}} + 1 - \beta_1.$$

$$\lambda_{t+1} = \frac{x_{t+1}^{k-2}}{\sqrt{v_{t+1}}} = \frac{(1-\eta\omega_t\lambda_t)^{k-2}x_t^{k-2}}{\sqrt{\beta_2 v_t + (1-\beta_2)(1-\eta\omega_t\lambda_t)^{2k-2}x_t^{2k-2}}} = \frac{(1-\eta\omega_t\lambda_t)^{k-2}\lambda_t}{\sqrt{\beta_2 + (1-\beta_2)(1-\eta\omega_t\lambda_t)^{2k-2}\lambda_t^2 x_t^2}}.$$

So we have the following dynamics:

$$\omega_{t+1} = \frac{\beta_1\omega_t}{(1-\eta\omega_t\lambda_t)^{k-1}} + 1 - \beta_1$$

$$\lambda_{t+1} = \frac{(1-\eta\omega_t\lambda_t)^{k-2}\lambda_t}{\sqrt{\beta_2 + (1-\beta_2)(1-\eta\omega_t\lambda_t)^{2k-2}\lambda_t^2 x_t^2}}$$

$$x_{t+1} = (1 - \eta\omega_t\lambda_t)x_t \tag{27}$$

Let $(\omega^*, \lambda^*, x^*)$ a fixed point of the dynamics (27). Then we have

$$x^* = (1 - \eta\omega^*\lambda^*x^*).$$

So $x^*\omega^*\lambda^* = 0$. If $x^* = 0$, by Equation (27), we have

$$\omega^* = \frac{\beta_1\omega^*}{(1-\eta\omega^*\lambda^*)^{k-1}} + 1 - \beta_1$$

$$\lambda^* = \frac{(1-\eta\omega^*\lambda^*)^{k-2}\lambda^*}{\sqrt{\beta_2}}$$

Solve this equations, and we get

$$\omega^* = 1, \lambda^* = 0 \quad \text{or} \quad \omega^* = \frac{1-\beta_1}{1 - \beta_1\beta_2^{-\frac{k-1}{2(k-2)}}}, \lambda^* = \frac{1 - \beta_2^{\frac{1}{2(k-2)}}}{\eta\omega^*}$$

If $x^* \neq 0$, then $\omega^*\lambda^* = 0$. So

$$\omega^* = \beta_1\omega^* + (1-\beta_1).$$

So we have

$$\omega^* = 1.$$

Since $\omega^*\lambda^* = 0$, so we have $\lambda^* = 0$. It is readily to verify that if $\omega^* = 1, \lambda^* = 0$, then $(\omega^*, \lambda^*, x)$ is fixed point of (27) for any $x \in \mathbb{R}$.

Therefore, the fixed point $(\omega^*, \lambda^*, x^*)$ of (27) has two cases:

- $\omega^* = \frac{1-\beta_1}{1-\beta_1\beta_2^{-\frac{k-1}{2(k-2)}}}, \lambda^* = \frac{1-\beta_2^{\frac{1}{2(k-2)}}}{\eta\omega^*}, x^* = 0$
- $\omega^* = 1, \lambda^* = 0, x^* \in \mathbb{R}$

Besides, since $\lambda_t = \frac{x_t^{k-2}}{\sqrt{v_t}}$, we have $\lambda_t \geq 0$. Therefore, if $\lambda^*$ is a limit point of $\lambda_t$, then $\lambda^* \geq 0$. Under this condition, the fix

point $\omega^* = \frac{1-\beta_1}{1-\beta_1\beta_2^{-\frac{k-1}{2(k-2)}}}, \lambda^* = \frac{1-\beta_2^{\frac{1}{2(k-2)}}}{\eta\omega^*}, x^* = 0$ exists if and only if $\omega^* > 0$, which is equivalent to

$$\beta_2^{\frac{k-1}{2(k-2)}} > \beta_1.$$

By calculation, the Jacobian matrix at the fixed point $\omega^* = 1, \lambda^* = 0, x^* \in \mathbb{R}$ is

$$M = \begin{pmatrix} 1 & (k-1)\eta\beta_1 & 0 \\ 0 & \frac{1}{\sqrt{\beta_2}} & 0 \\ 0 & -\eta x^* & 1 \end{pmatrix}.$$

The eigenvalues of $M$ are $1, 1, \frac{1}{\sqrt{\beta_2}}$. Since $\frac{1}{\sqrt{\beta_2}} > 1$, the fixed point $\omega^* = 1, \lambda^* = 0, x^*$ is unstable for each $x^* \in \mathbb{R}$. So far, we have proved the first four statements of the theorem.

**Proof of Statements 5-6:**

By calculation, the Jacobian matrix at this fixed point is of the form

$$J' = \begin{bmatrix} J & 0 \\ 0 & \beta_2^{\frac{1}{2(k-2)}} \end{bmatrix},$$

where

$$J = \begin{bmatrix} \beta_1\left(1 - \beta_2^{\frac{1}{2(k-2)}}\right)(k-1)\beta_2^{\frac{-k}{2(k-2)}} + \beta_1\beta_2^{\frac{1-k}{2(k-2)}} & \dfrac{\beta_1\eta(1-\beta_1)^2(k-1)\beta_2^{\frac{-k}{2(k-2)}}}{\left(-\beta_1\beta_2^{-\frac{k-1}{2(k-2)}}+1\right)^2} \\[2em] \dfrac{\left(1 - \beta_2^{\frac{1}{2(k-2)}}\right)^2(2-k)\left(-\beta_1\beta_2^{-\frac{k-1}{2(k-2)}}+1\right)^2\beta_2^{\frac{k-3}{2(k-2)}}}{\sqrt{\beta_2}\eta(1-\beta_1)^2} & \dfrac{\left(1 - \beta_2^{\frac{1}{2(k-2)}}\right)(2-k)\beta_2^{\frac{k-3}{2(k-2)}}+\sqrt{\beta_2}}{\sqrt{\beta_2}} \end{bmatrix}$$

Since $\beta_2^{\frac{1}{2(k-2)}} \in (0,1)$, then we have

$$r(J') < 1 \iff r(J) < 1.$$

Let $r(J)$ be the spectral radius of $J$. Note that $r(J) < 1$ if and only if all of the following satisfy:

- $|\det(J)| < 1$.
- $1 - \text{tr}(J) + \det(J) > 0$
- $1 + \text{tr}(J) + \det(J) > 0$

Besides, $r(J) = 1$ if and only if at least one of the inequalities becomes an equality.

By calculation, we have

$$\det(J) = \beta_1\beta_2^{-\frac{k}{2(k-2)}}$$

$$1 - \text{tr}(J) + \det(J) = (2-k)(1 - \beta_2^{\frac{1}{2(k-2)}})(\beta_1\beta_2^{\frac{-k}{2(k-2)}} - \beta_2^{-\frac{1}{2(k-2)}}).$$

Therefore, we have

$$|\det(J)| < 1 \implies 1 - \text{tr}(J) + \det(J) > 0.$$

By calculation, the condition $1 + \text{tr}(J) + \det(J) > 0$ is simplified to

$$\beta_1 > \frac{(k-2)\beta_2^{-\frac{1}{2(k-2)}} - k}{(k - (k-2)\beta_2^{\frac{1}{2(k-2)}})\beta_2^{-\frac{k}{2(k-2)}}}.$$

So $r(J) < 1$ if and only if both of the two inequalities hold:

$$\beta_1 < \beta_2^{\frac{k}{2(k-2)}}$$

$$\beta_1 > \frac{(k-2)\beta_2^{-\frac{1}{2(k-2)}} - k}{(k - (k-2)\beta_2^{\frac{1}{2(k-2)}})\beta_2^{-\frac{k}{2(k-2)}}}.$$

Besides, $r(J) = 1$ if and only if at least one of these two inequalities becomes an equality.

Moreover, one may verify that when $\beta_2 > (\frac{k-2}{k})^{2(k-2)}$, we have $J_{11} > 0$, $J_{22} > -1$. So we have

$$1 + \text{tr}(J) + \det(J) > 0.$$

Therefore, when $\beta_2 > (\frac{k-2}{k})^{2(k-2)}$, $r(J) < 1$ if and only if $\beta_1 < \beta_2^{\frac{k}{2(k-2)}}$. So far, we have proved the first six statements of the theorem.

**Proof of Statements 7:**

Assume that $\beta_1 < \beta_2^{\frac{k}{2(k-2)}}$ and $\beta_1 > \frac{(k-2)\beta_2^{-\frac{1}{2(k-2)}} - k}{(k-(k-2)\beta_2^{\frac{1}{2(k-2)}})\beta_2^{-\frac{k}{2(k-2)}}}$, and consider the fixed point $\omega^* = \frac{1-\beta_1}{1-\beta_1\beta_2^{-\frac{k-1}{2(k-2)}}}, \lambda^* = \frac{1-\beta_2^{\frac{1}{2(k-2)}}}{\eta\omega^*}, x^* = 0$. The Jacobian matrix at this fixed point is

$$J' = \begin{bmatrix} J & 0 \\ 0 & \beta_2^{\frac{1}{2(k-2)}} \end{bmatrix}.$$

By statements 6, we have $r(J) < 1$. Since $\beta_2^{\frac{1}{2(k-2)}} \in (0,1)$, so we have

$$r(J') < 1.$$

This indicates that this fixed point is asymptotically stable. By the definition of asymptotical stability, for the initialization $(\omega_0, \lambda_0, x_0)$ sufficiently close to the fixed point, the iterations $(\omega_t, \lambda_t, x_t)$ will converge to $(\omega^*, \lambda^*, x^*)$.

By definition, we have

$$\frac{x_{t+1}}{x_t} = 1 - \eta\omega_t\lambda_t.$$

So we have

$$\lim_{t\to\infty} \frac{x_{t+1}}{x_t} = 1 - \eta\omega^*\lambda^* = \beta_2^{\frac{1}{2(k-2)}}.$$

**Proof of Statements 8:** When $\beta_1 > \beta_2^{\frac{k}{2(k-2)}}$ or $\beta_1 < \frac{(k-2)\beta_2^{-\frac{1}{2(k-2)}} - k}{(k-(k-2)\beta_2^{\frac{1}{2(k-2)}})\beta_2^{-\frac{k}{2(k-2)}}}$, the Jacobian matrix at the fixed point

$\omega^* = \frac{1-\beta_1}{1-\beta_1\beta_2^{-\frac{k-1}{2(k-2)}}}, \lambda^* = \frac{1-\beta_2^{\frac{1}{2(k-2)}}}{\eta\omega^*}, x^* = 0$ has a spectral radius larger than one. So the fixed point is unstable.

$\square$

## B. Proof of Global Convergence of RMSProp Theorem 5.7

$L(x) = \frac{1}{k}x^k$, $k$ is an even number and $k \geq 4$.

$$x_{t+1} = x_t - \eta \cdot \frac{x_t^{k-1}}{\sqrt{v_t}}$$

$$v_{t+1} = \beta_2 v_t + (1-\beta_2)x_{t+1}^{2k-2} \tag{28}$$

**Lemma B.1.** *If* $\eta\frac{x_0^{k-2}}{\sqrt{v_0}} < 1$ *and* $\frac{(k-2)^{k-2}}{(k-1)^{k-1}} < \sqrt{\beta_2} < 1$*, then there exists* $T > 0$ *such that* $|x_t|$ *of Equation 28 decreases monotonically for* $t > T$*. Moreover,* $x_t$ *converges to* $0$*.*

*Proof.* Without loss of generality assume $x_0 > 0$. We define

$$\lambda_t = \frac{x_t^{k-2}}{\sqrt{v_t}},$$

and then we have the following dynamics:

$$x_{t+1} = (1 - \eta\lambda_t)x_t$$

$$\lambda_{t+1} = \frac{(1 - \eta\lambda_t)^{k-2}\lambda_t}{\sqrt{\beta_2 + (1 - \beta_2)(1 - \eta\lambda_t)^{2k-2}x_t^2\lambda_t^2}} \quad (29)$$

So we have the following inequality:

$$0 < \lambda_{t+1} < \frac{(1 - \eta\lambda_t)^{k-2}\lambda_t}{\sqrt{\beta_2}}.$$

By assumption, we have $\lambda_0 < \frac{1}{\eta}$. Define $g(\lambda) = \frac{1}{\sqrt{\beta_2}}(1 - \eta\lambda)^{k-2}\lambda$, $\lambda \in (0, \frac{1}{\eta})$. By calculation, the maximum of $g(\lambda)$ is

$$g(\frac{1}{(k-1)\eta}) = \frac{(k-2)^{k-2}}{(k-1)^{k-1}\eta\sqrt{\beta_2}}.$$

If $\sqrt{\beta_2} > \frac{(k-2)^{k-2}}{(k-1)^{k-1}}$, then $0 < 1 - \eta\lambda_{t+1} < 1$. So $x_t$ decreases monotonically. Since $x_t > 0$, so $x_t$ converges to some $x^* \geq 0$. In the equation

$$x_{t+1} = x_t - \eta \cdot \frac{x_t^{k-1}}{\sqrt{v_t}},$$

take limsup and liminf and we get

$$\frac{x^*}{\sqrt{\liminf_{t\to\infty} v_t}} = \frac{x^*}{\sqrt{\limsup_{t\to\infty} v_t}} = 0.$$

If $x^* \neq 0$, then $\lim_{t\to\infty} v_t = +\infty$. This is impossible since

$$v_{t+1} = \beta_2 v_t + (1 - \beta_2)x_{t+1}^{2k-2}$$

and $\beta_2 < 1$. So $x^*$ must be zero.

$\square$

**Lemma B.2.** *Assume* $0 < \lambda_0 < \frac{1}{\eta}$. *Assume* $\frac{(k-2)^{k-2}}{k^{k-2}} < \sqrt{\beta_2} < 1$. *Then* $\lambda_t$ *of Equation* (29) *converges to* $\frac{1-\beta_2^{\frac{1}{2(k-2)}}}{\eta}$.

*Proof.* Define $g(\lambda) = \frac{1}{\sqrt{\beta_2}}(1 - \eta\lambda)^{k-2}\lambda$, $\lambda \in (0, \frac{1}{\eta})$. The fixed point of $g$ in $[0, \frac{1}{\eta}]$ is

$$\lambda^* = \frac{1 - \beta_2^{\frac{1}{2(k-2)}}}{\eta} \quad \text{and} \quad \lambda' = 0.$$

The fixed point $\lambda' = 0$ is unstable. The derivative at $\lambda^* = \frac{1-\beta_2^{\frac{1}{2(k-2)}}}{\eta}$ is

$$g'(\lambda^*) = 1 - (k-2)(\beta_2^{-\frac{1}{2(k-2)}} - 1).$$

Then $g'(\lambda^*) < 1$. It is readily to verity that

$$g'(\lambda^*) > -1 \iff \sqrt{\beta_2} > \frac{(k-2)^{k-2}}{k^{k-2}}.$$

By assumption, $\sqrt{\beta_2} > \frac{(k-2)^{k-2}}{k^{k-2}}$ holds. Therefore, we have

$$g'(\lambda^*) \in (-1, 1).$$

So $\lambda^*$ is asymptotically stable.

We first present a lemma along with its proof:

**Lemma B.3.** *If $\frac{(k-2)^{k-2}}{k^{k-2}} < \sqrt{\beta_2} < 1$, then the dynamical system defined by $g(s_{t+1}) = s_t$, with initial condition $s_0 \in \left(0, \frac{1}{\eta}\right)$, globally converges to the fixed point $s^* = \frac{1 - \beta_2^{\frac{1}{2k-4}}}{\eta}$.*

*Proof.* By definition, we have $g(s) = \frac{1}{\sqrt{\beta_2}}(1 - \eta s)^{k-2}s, s_0 \in (0, \frac{1}{\eta})$. Then we have

$$g(\eta s) = \frac{1}{\sqrt{\beta_2}}(1 - \eta s)^{k-2}\eta s, \eta s_0 \in (0, 1).$$

Therefore, we only need to study the case of $\eta = 1$. In this case, we have $g(s) = \frac{1}{\sqrt{\beta_2}}(1-s)^{k-2}s, s_0 \in (0, 1)$.

Consider the roots of $g(g(s)) = s$. It is readily to verify its the roots in $[0, 1]$ are all fixed points.

By Coppel (1955), the dynamics $g(s_{t+1}) = s_t$ with $s_0 \in (0, 1)$ must converge to a fixed point of $g$. By calculation, there are two fixed points of $g$:

$$s^1 = 0, s^2 = 1 - \beta_2^{\frac{1}{2k-4}}.$$

By calculation, $g'(0) = \frac{1}{\sqrt{\beta_2}} > 1$. So the fixed point $s^1 = 0$ is repelling. So $s_t$ cannot converge to zero. So $s_t$ must converge to $s^2 = 1 - \beta_2^{\frac{1}{2k-4}}$. This proves the lemma.

$\square$

We continue our proof of Lemma B.2. It is easy to verify that $k^{k-2} < (k-1)^{k-1}$. So the assumptions of Lemma B.1 holds. By Lemma B.1, we have

$$\lim_{t \to \infty} x_t = 0.$$

So the dynamics of $\lambda_t$ is

$$\lambda_{t+1} = g(\lambda_t) + \mu(t), \text{ where } \lim_{t \to \infty} \mu(t) = 0.$$

We now consider the limit point of $\lambda_t$. Let $\lambda^l$ be a limit point of $\lambda_t$. Define $g_{\max} = \max_{\lambda \in (0, \frac{1}{\eta})} g(\lambda)$. It is readily to verify that $0 < g_{\max} < \frac{1}{\eta}$. Since $\lambda^l$ be a limit point of $\lambda_t$, we have

$$0 \le \lambda^l \le g_{\max} < \frac{1}{\eta}.$$

- We first prove that $\lambda^l \neq 0$. Assume by contradiction that $\lambda_l = 0$. Since $g'(0) > 1$, $g(\lambda) < g_{\max} < \frac{1}{\eta}$ and $\lim_{t \to \infty} \epsilon(t) = 0$, the following holds:

$$\exists T_0 > 0, \delta > 0, 0 < c < 1 \quad s.t. \quad \text{for all } t > T_0, \text{ if } \lambda_t < \delta, \text{ then } \lambda_{t-1} < c\lambda_t.$$

Since $\lambda^l = 0$, there exists $T_n > T_0$, $\lim_{n \to \infty} T_n = +\infty$, such that $\lambda_{T_n} < \delta$. So

$$\lambda_{T_0} < \lambda_{T_n} c^{T_n - T_0} < \frac{1}{\eta} c^{T_n - T_0}.$$

Let $n \to \infty$, we get $\lambda_{T_0} = 0$. This is impossible when $\lambda_0 \in (0, \frac{1}{\eta})$.

- Then we prove that the existence of $\lambda^l$ such that $\lambda^l \neq 0$ and $\lambda^l \neq \lambda^*$ will lead to a contradiction. Pick any $\epsilon > 0$. Consider the reference dynamics given by

$$s_{t+1} = g(s_t), s_0 = s_0$$

and denotes its solution by $\phi(t, s_0)$. By Lemma B.3, $\phi(t, \lambda^l)$ converges to $\lambda^*$. Since $\phi(t, \lambda^l)$ converges to $\lambda^*$, there exists $T > 0$ such that

$$\phi(T, \lambda^l) \in (\lambda^* - \frac{1}{4}\epsilon, \lambda^* + \frac{1}{4}\epsilon).$$

Moreover, since $\phi(t, s)$ is continuously over $s$, there exists $\delta > 0$ such that

$$\phi(T, s_0) \in (\lambda^* - \frac{1}{2}\epsilon, \lambda^* + \frac{1}{2}\epsilon)$$

for all $s_0 \in (\lambda^l - \delta, \lambda^l + \delta)$. Since $\lambda^l$ is a limit point of $\lambda_t$, there exists $T_k \to \infty$ such that

$$\lambda_{T_k} \in (\lambda^l - \delta, \lambda^l + \delta), \forall k \in \mathbb{N}^+.$$

Since $\lambda_{T_k} \in (\lambda^l - \delta, \lambda^l + \delta)$, we have

$$\phi(T, \lambda_{T_k}) \in (\lambda^* - \frac{1}{2}\epsilon, \lambda^* + \frac{1}{2}\epsilon).$$

Denote the solution of

$$w_{t+1} = g(w_t) + h(t), w_0 = w_0$$

by $\Phi_h(t, w_0)$. By continuity of $\Phi$ with respect to $h(\cdot)$ and the initialization, there exists $\delta' > 0$ such that if $|\mu(t)| < \delta'$ for all $t \in [0, T]$ and $|\lambda - \lambda^l| < \delta'$, then

$$|\Phi_h(T, \lambda) - \phi(T, \lambda^l)| < \epsilon. \tag{30}$$

We may pick $\delta < \delta'$. Then we have

$$|\lambda_{T_k} - \lambda^l| < \delta'.$$

Since $\lim_{t \to \infty} \mu(t) = 0$ and $T_k \to \infty$, we may assume that $|\mu(t + T_k)| < \epsilon$. In inequality (30), let $h(t) = \mu(t + T_k)$ and $\lambda = \lambda_{T_k}$, we get

$$|\lambda_{T+T_k} - \phi(T, \lambda^l)| < \epsilon.$$

Since

$$\phi(T, \lambda^l) \in (\lambda^* - \frac{1}{4}\epsilon, \lambda^* + \frac{1}{4}\epsilon),$$

we have

$$\lambda_{T+T_k} \in (\lambda^* - 2\epsilon, \lambda^* + 2\epsilon), \forall k \in \mathbb{N}^+. \tag{31}$$

In the next we prove (31) leads to a contradiction.
Since $|g'(\lambda^*)| < 1$, there exists $\delta > 0$, such that

$$|g'(\lambda)| < d < 1 \text{ for all } \lambda \in (\lambda^* - \delta, \lambda^* + \delta)$$

Since $\lim_{t \to \infty} \epsilon(t) = 0$ and Equation (31), for any $\epsilon > 0$, there exists $T$ such that

$$\lambda_T \in (\lambda^* - \frac{1}{2}\delta, \lambda^* + \frac{1}{2}\delta) \text{ and } \epsilon(t) < \epsilon, \forall t > T.$$

When $\lambda_t \in (\lambda^* - \delta, \lambda^* + \delta)$, we have

$$|\lambda_{t+1} - \lambda^*| = |g(\lambda_t) - g(\lambda^*) - \epsilon(t)| \le d|\lambda_t - \lambda^*| + \epsilon. \tag{32}$$

Since $\lambda_T \in (\lambda^* - \frac{1}{2}\delta, \lambda^* + \frac{1}{2}\delta)$, we may pick $\epsilon$ sufficiently small such that

$$\lambda_t \in (\lambda^* - \delta, \lambda^* + \delta), \forall t > T.$$

Then Equation (32) holds for all $t > T$.
Moreover, since $d < 1$, there exists $T' > 0$ such that

$$|\lambda_t - \lambda^*| < 2\epsilon, \forall t > T'.$$

This is the definition of the limit

$$\lim_{t \to \infty} \lambda_t = \lambda^*.$$

So $\lambda^l \ne \lambda^*$ leads to a contradiction.

Therefore $\lambda_t$ cannot has any limit point except than $\lambda^*$. So $\lambda_t$ converges to $\lambda^*$.

$\square$

**Proof of Theorem 5.7:**

*Proof.* By Lemma B.1, $x_t$ converges to zero. By Lemma B.2, $\lambda_t$ converges to $\frac{1-\beta_2^{\frac{1}{2(k-2)}}}{\eta}$. By definition, we have

$$\frac{x_{t+1}}{x_t} = 1 - \eta\lambda_t.$$

Take the limit, one get

$$\lim_{t\to\infty} \frac{x_{t+1}}{x_t} = \beta_2^{\frac{1}{2(k-2)}}.$$

$\square$

## C. Convergence Rate of GD and Momentum on Degenerated Polynomials

**Theorem C.1** (**Power-Law Convergence of Gradient Flow**). *Consider the gradient flow dynamics on $L(x) = \frac{1}{k}x^k$ with $k \geq 4$ and is even:*

$$\frac{dx}{dt} = -\eta\nabla L(x) = -\eta x^{k-1}, \quad x(0) = x_0 \neq 0. \tag{33}$$

*For any learning rate $\eta > 0$, the solution $x(t)$ exhibits an asymptotic power-law decay:*

$$x(t) = \Theta\left(t^{-\frac{1}{k-2}}\right) \quad \text{as } t \to \infty. \tag{34}$$

*Proof.* Without loss of generality, assume $x_0 > 0$. The dynamics remain in the positive domain, allowing us to write $\dot{x} = -\eta x^{k-1}$. We employ the separation of variables method:

$$\frac{dx}{x^{k-1}} = -\eta \, dt. \tag{35}$$

Integrating from time $0$ to $t$:

$$\int_{x_0}^{x(t)} y^{-(k-1)} \, dy = -\eta \int_0^t ds. \tag{36}$$

Since $k \geq 4$, we have $k - 2 > 0$. Evaluating the integral yields:

$$\left[\frac{y^{-(k-2)}}{-(k-2)}\right]_{x_0}^{x(t)} = -\frac{1}{k-2}\left(x(t)^{-(k-2)} - x_0^{-(k-2)}\right) = -\eta t. \tag{37}$$

Rearranging terms to solve for $x(t)$:

$$x(t)^{-(k-2)} = x_0^{-(k-2)} + (k-2)\eta t. \tag{38}$$

Taking the power of $-\frac{1}{k-2}$, we obtain the explicit solution:

$$x(t) = \left[x_0^{-(k-2)} + (k-2)\eta t\right]^{-\frac{1}{k-2}}. \tag{39}$$

As $t \to \infty$, the term $(k-2)\eta t$ dominates the initial condition $x_0^{-(k-2)}$. Thus, the asymptotic behavior is given by:

$$x(t) \sim [(k-2)\eta t]^{-\frac{1}{k-2}} = \Theta(t^{-\frac{1}{k-2}}). \tag{40}$$

This confirms that the convergence rate is algebraic dependent on the degeneracy order $k$. $\square$

**Theorem C.2** (**Power-Law Convergence of Momentum**). *Consider the continuous-time Heavy-ball momentum dynamics with momentum factor $\beta_1 \in [0, 1)$:*

$$\dot{m} = -(1-\beta_1)m + (1-\beta_1)\nabla L(x), \quad m(0) = 0, \tag{41}$$

$$\dot{x} = -\eta m. \tag{42}$$

*For $L(x) = \frac{1}{k}x^k$ with $k \geq 4$ and is even, the trajectory $x(t)$ satisfies:*

$$x(t) = \Theta\left(t^{-\frac{1}{k-2}}\right) \quad as\ t \to \infty. \tag{43}$$

*Thus, momentum fails to improve the power-law exponent of Gradient Descent.*

*Proof.* We analyze the system by reducing it to a second-order differential equation. Differentiating (42) with respect to time gives $\ddot{x} = -\eta\dot{m}$. Substituting $\dot{m}$ from (41):

$$\ddot{x} = -\eta\left[-(1 - \beta_1)m + (1 - \beta_1)\nabla L(x)\right]. \tag{44}$$

Using $m = -\frac{1}{\eta}\dot{x}$, we obtain the damped nonlinear oscillator equation:

$$\underbrace{\ddot{x}}_{\text{Inertia}} + \underbrace{(1 - \beta_1)\dot{x}}_{\text{Friction}} + \underbrace{\eta(1 - \beta_1)x^{k-1}}_{\text{Gradient Force}} = 0. \tag{45}$$

To rigorously determine the asymptotic behavior as $t \to \infty$, we employ the *Method of Dominant Balance*. Since $\ddot{x}$ and $\dot{x}$ differ by a time derivative, they inherently scale differently for any asymptotically vanishing polynomial solution, precluding a simultaneous three-term balance. Thus, we systematically evaluate all three possible pairwise balances among the terms in Eq. (45) to identify the uniquely consistent asymptotic regime. Because the friction coefficient $(1 - \beta_1)$ is always greater than zero, the system's energy will eventually decay to zero. Thus we analyze the asymptotic behavior as $x \to 0$.

### Balance 1: Inertia $\sim$ Friction $\gg$ Gradient.
Assume the system is dominated by inertia and friction, such that $\ddot{x} \sim -(1 - \beta_1)\dot{x}$. The solution to this linear ODE is $x(t) \sim C_1 + C_2e^{-(1-\beta_1)t}$. For the trajectory to converge to the minimum $x = 0$, we must have $C_1 = 0$, yielding $x(t) \sim C_2e^{-(1-\beta_1)t}$. We must check if the neglected Gradient term is indeed smaller than the residual of the dominant terms. Substituting $x(t)$ into the full equation, the residual of the dominant terms is exactly $0$ (since it is the homogeneous solution). However, the neglected Gradient term scales as $x^{k-1} \sim e^{-(k-1)(1-\beta_1)t} \neq 0$. Since the non-zero gradient force continuously drives the system towards the minimum, it cannot be balanced by a zero residual. Physically, this exponential behavior only captures the transient "coasting" phase, not the global asymptotic convergence driven by the polynomial basin. Thus, this balance is invalid as $t \to \infty$.

### Balance 2: Inertia $\sim$ Gradient $\gg$ Friction.
Assume the system is dominated by inertia and the gradient, i.e., $\ddot{x} \sim -\eta(1 - \beta_1)x^{k-1}$. This represents a conservative, undamped oscillator. Multiplying by $\dot{x}$ and integrating gives the energy conservation $\frac{1}{2}\dot{x}^2 + \frac{\eta(1-\beta_1)}{k}x^k \approx E$. As $x \to 0$, the characteristic velocity scales as $\dot{x} \sim x^{k/2}$. We now check the neglected Friction term: Friction $\sim \dot{x} \sim x^{k/2}$. By our assumption, Friction must be strictly smaller than the Gradient: $x^{k/2} \ll x^{k-1}$. Since $x \to 0$ asymptotically, this requires $k/2 > k - 1$, which implies $k < 2$. However, we are given $k \geq 4$. Thus, Friction actually dominates the Gradient, directly contradicting the assumption. This balance is invalid.

### Balance 3: Friction $\sim$ Gradient $\gg$ Inertia.
Assume the system is heavily damped, where friction balances the gradient:

$$(1 - \beta_1)\dot{x} \sim -\eta(1 - \beta_1)x^{k-1} \implies \dot{x} \sim -\eta x^{k-1}. \tag{46}$$

By solving this first-order ODE, we obtain the asymptotic decay rate $x(t) \sim [(k-2)\eta t]^{-\frac{1}{k-2}}$, yielding $x(t) = \Theta\left(t^{-\frac{1}{k-2}}\right)$. We must verify that the neglected Inertia term is indeed asymptotically smaller than Friction. Differentiating the solution:

$$\dot{x}(t) \sim t^{-\left(\frac{1}{k-2}+1\right)} = t^{-\frac{k-1}{k-2}}, \tag{47}$$

$$\ddot{x}(t) \sim t^{-\left(\frac{k-1}{k-2}+1\right)} = t^{-\frac{2k-3}{k-2}}. \tag{48}$$

We require $|\ddot{x}| \ll |\dot{x}|$ as $t \to \infty$. Comparing their powers:

$$-\frac{2k-3}{k-2} < -\frac{k-1}{k-2} \iff 2k - 3 > k - 1 \iff k > 2. \tag{49}$$

Since $k \geq 4$, the condition $k > 2$ is strictly satisfied. The inertia term decays as $O(t^{-1})$ relative to the friction and gradient terms:

$$\frac{|\ddot{x}|}{|\dot{x}|} \sim \frac{t^{-\frac{2k-3}{k-2}}}{t^{-\frac{k-1}{k-2}}} = t^{-1} \to 0 \quad \text{as } t \to \infty. \tag{50}$$

**Conclusion.**
Out of all possible asymptotic balances, only the over-damped regime (Friction $\sim$ Gradient) is mathematically self-consistent for $k \geq 4$. Therefore, the continuous-time momentum dynamics are asymptotically governed by the first-order gradient flow, and the trajectory strictly follows $x(t) = \Theta\left(t^{-\frac{1}{k-2}}\right)$. $\square$

**Lemma C.3** (**Exponential Convergence via Exponential Learning Rate Schedule**). *Under the setup of the degenerate objective $L(x) = \frac{1}{k}x^k$ ($k \geq 4$, even), consider the gradient flow dynamics with a time-varying learning rate $\eta(t)$:*

$$\frac{dx}{dt} = -\eta(t)\nabla L(x) = -\eta(t)x^{k-1}. \tag{51}$$

*If the learning rate follows an exponential schedule $\eta(t) = \eta_0 e^{\alpha t}$ with $\alpha > 0$ and $\eta_0 > 0$, then the solution $x(t)$ exhibits asymptotic exponential convergence:*

$$x(t) \sim C \exp\left(-\frac{\alpha}{k-2}t\right) \quad \text{as } t \to \infty, \tag{52}$$

*where $C = \left[\frac{(k-2)\eta_0}{\alpha}\right]^{-\frac{1}{k-2}}$ is a constant determined by the schedule parameters.*

*Proof.* We employ the separation of variables method. Rewriting the differential equation:

$$\frac{dx}{x^{k-1}} = -\eta_0 e^{\alpha t} dt. \tag{53}$$

Integrating both sides from the initial state $0$ to time $t$:

$$\int_{x_0}^{x(t)} y^{-(k-1)} dy = -\eta_0 \int_0^t e^{\alpha s} ds. \tag{54}$$

Computing the definite integrals:

$$\left[\frac{y^{-(k-2)}}{-(k-2)}\right]_{x_0}^{x(t)} = -\frac{\eta_0}{\alpha}(e^{\alpha t} - 1). \tag{55}$$

This simplifies to:

$$\frac{x(t)^{-(k-2)} - x_0^{-(k-2)}}{-(k-2)} = -\frac{\eta_0}{\alpha}(e^{\alpha t} - 1). \tag{56}$$

Multiplying by $-(k-2)$ and rearranging for the dominant term:

$$x(t)^{-(k-2)} = x_0^{-(k-2)} + \frac{(k-2)\eta_0}{\alpha}(e^{\alpha t} - 1). \tag{57}$$

As $t \to \infty$, the term $e^{\alpha t}$ dominates both the constant $-1$ and the initial condition term $x_0^{-(k-2)}$. Thus, we have the asymptotic relation:

$$x(t)^{-(k-2)} \approx \frac{(k-2)\eta_0}{\alpha} e^{\alpha t}. \tag{58}$$

Taking the power of $-\frac{1}{k-2}$ on both sides yields the solution:

$$x(t) \approx \left[\frac{(k-2)\eta_0}{\alpha}\right]^{-\frac{1}{k-2}} \exp\left(-\frac{\alpha}{k-2}t\right). \tag{59}$$

This confirms that the convergence rate is linear (exponential in time) with rate constant $\lambda = \frac{\alpha}{k-2}$. $\square$

# D. Proof of Proposition 5.6

**Proposition D.1** (**Stability of the Acceleration Map**). *The map in Eq.* (18) *has a non-zero fixed point* $u^* = 1 - \gamma^{-\frac{1}{k-2}} \in (0,1)$, *locally asymptotically stable if and only if:*

$$\gamma < \gamma_{crit} := \left(\frac{k}{k-2}\right)^{k-2}. \tag{60}$$

*Proof.* We provide the detailed derivation for the existence and linear stability of the fixed point for the effective sharpness map defined in Eq. (18).

Consider the gradient descent update rule for the degenerate loss function $L(x) = \frac{1}{k}x^k$ ($k \geq 4$ and even) with an exponentially increasing learning rate sequence:

$$x_{t+1} = x_t - \eta_t x_t^{k-1} = (1 - \eta_t x_t^{k-2})x_t, \tag{61}$$

$$\eta_{t+1} = \gamma\eta_t, \quad (\gamma > 1). \tag{62}$$

**Reduction to a One-Dimensional Map.** To analyze the long-term behavior of this coupled system, we introduce a dimensionless invariant, the *effective sharpness*, denoted as $u_t$. This variable captures the interplay between the learning rate and the local Hessian geometry ($L''(x) \propto x^{k-2}$):

$$u_t := \eta_t x_t^{k-2}. \tag{63}$$

The dynamics of the optimization process are fully characterized by the evolution of $u_t$. In the continuous limit (Lem. 5.5), the product $\eta(t)x(t)^{k-2}$ remains $O(1)$, maintaining stable exponential convergence. However, in the discrete setting, stability depends on whether $u_t$ remains within the "contractive domain" $(0, 2)$. If $u_t > 2$, the step size exceeds the stability limit of the local curvature, leading to divergence.

Substituting (61) and (62) into the definition of $u_t$, we derive the recurrence relation:

$$u_{t+1} = \eta_{t+1}(x_{t+1})^{k-2} = (\gamma\eta_t) \cdot \left[x_t(1 - \eta_t x_t^{k-2})\right]^{k-2}$$
$$= \gamma(\eta_t x_t^{k-2})(1 - \eta_t x_t^{k-2})^{k-2}.$$

This transformation decouples the analysis from explicit time dependence, reducing the high-dimensional system $(x, \eta)$ to a one-dimensional fixed-point iteration map:

$$\boxed{u_{t+1} = f(u_t) = \gamma u_t(1 - u_t)^{k-2}.} \tag{64}$$

We now analyze the dynamical properties of this discrete map. First, $0$ is a trivial fixed point, which can be easily shown unstable.

**i. Existence of the non-trivial Fixed Point.** Let $u^*$ denote a non-zero fixed point of the map (64). Setting $u_{t+1} = u_t = u^*$, we obtain:

$$1 = \gamma(1 - u^*)^{k-2} \implies u^* = 1 - \gamma^{-\frac{1}{k-2}}. \tag{65}$$

For $\gamma > 1$, we have $u^* \in (0, 1)$. This implies that at equilibrium, the contraction factor $(1 - u^*)$ lies in $(0, 1)$, ensuring the monotonic convergence of the parameter $x_t$.

**ii. Linear Stability Analysis.** The fixed point $u^*$ is locally asymptotically stable if and only if the spectral radius of the map's Jacobian at $u^*$ is less than unity:

$$|f'(u^*)| < 1. \tag{66}$$

Differentiating $f(u)$ with respect to $u$:

$$f'(u) = \gamma\left[(1 - u)^{k-2} - u(k-2)(1 - u)^{k-3}\right]$$
$$= \gamma(1 - u)^{k-3}\left[1 - (k-1)u\right]. \tag{67}$$

Evaluating at $u^*$ and utilizing the identity $\gamma(1 - u^*)^{k-2} = 1$, we simplify the expression:

$$f'(u^*) = \frac{1}{1 - u^*}[1 - (k - 1)u^*]. \tag{68}$$

Let $\mu = \gamma^{-\frac{1}{k-2}} = 1 - u^*$. Substituting $\mu$ yields:

$$f'(u^*) = \frac{1}{\mu}[1 - (k - 1)(1 - \mu)] = \frac{2 - k}{\mu} + (k - 1). \tag{69}$$

**iii. The Stability Threshold.** Given $k \geq 4$ and $\mu \in (0, 1)$, the upper bound condition $f'(u^*) < 1$ simplifies to $\frac{2-k}{\mu} + k - 2 < 0$, which is automatically satisfied. The stability is thus determined by the lower bound condition $f'(u^*) > -1$, which leads to:

$$\frac{2 - k}{\mu} + k - 1 > -1 \implies \frac{k - 2}{\mu} < k \implies \mu > \frac{k - 2}{k}.$$

Substituting $\mu = \gamma^{-\frac{1}{k-2}}$, we obtain the critical upper bound for the growth factor:

$$\gamma < \gamma_{\text{crit}} := \left(\frac{k}{k - 2}\right)^{k-2} = \left(1 + \frac{2}{k - 2}\right)^{k-2}. \tag{70}$$

$\square$

# E. Intuitive Understanding of Decoupling between Second Moment $v_t$ and Squared Gradients $g_t^2$

Given that exponential learning rate scaling is a prerequisite for linear convergence on degenerate landscapes (as established in Lem. 5.5), we now investigate how adaptive algorithms naturally induce this scaling. The underlying mechanism relies on a dynamic *decoupling* between the second moment estimate $v_t$ and the instantaneous squared gradient $g_t^2$. Specifically, if the gradient signal decays faster than the memory decay rate of the optimizer, $v_t$ ceases to track $g_t^2$ and instead follows its own inertial dynamics.

To formalize this, we define the *coupling ratio*, $R_t^{(v)}$, which quantifies the relative magnitude of the optimizer's accumulated state to the instantaneous signal:

$$R_t^{(v)} := \frac{v_t}{g_t^2}. \tag{71}$$

The evolution of this ratio determines whether the optimizer faithfully tracks the current landscape geometry ("Coupling") or is dominated by historical inertia ("Decoupling").

**The Dynamics of Coupling Ratio.** By substituting the update rule $v_t = \beta_2 v_{t-1} + (1 - \beta_2)g_t^2$ into the definition of $R_t^{(v)}$, we derive the following recurrence relation:

$$R_t^{(v)} = \beta_2 \frac{v_{t-1}}{g_t^2} + (1 - \beta_2)$$

$$= \beta_2 R_{t-1}^{(v)} \frac{g_{t-1}^2}{g_t^2} + (1 - \beta_2) := \left(\frac{\beta_2}{\rho_t}\right) R_{t-1}^{(v)} + (1 - \beta_2).$$

Here, $\rho_t := g_t^2/g_{t-1}^2$ represents the local decay rate of the squared gradient. The dynamical behavior of the optimizer is governed by the competition between the signal decay $\rho_t$ and the memory factor $\beta_2$:

- **Coupled Phase ($\rho_t > \beta_2$):** The gradient signal decays slowly relative to the memory fading. The recurrence is dominated by the source term $(1 - \beta_2)$, ensuring that the state estimate accurately tracks the instantaneous geometry ($v_t \approx g_t^2$).

- **Decoupled Phase ($\rho_t < \beta_2$):** The signal vanishes faster than the optimizer's forgetting rate. The inertial term $(\beta_2/\rho_t)R_{t-1}^{(v)}$ dominates (since $\beta_2/\rho_t > 1$), causing $v_t$ to disconnect from the gradient and enter a regime of autonomous exponential decay ($v_t \approx \beta_2 v_{t-1}$).

**Triggering the Exponential Stage.** Assuming standard initialization $v_0 = g_0^2$, $v_t$ is initially tightly coupled with $g_t^2$, thus the early training dynamics approximate SignGD with a constant step size $\eta$. Consequently, the gradient decay rate $\rho_t$ evolves as:

$$\rho_t = \left(\frac{x_t}{x_{t-1}}\right)^{2k-2} \approx \left(\frac{x_{t-1} - \eta}{x_{t-1}}\right)^{2k-2} = \left(1 - \frac{\eta}{x_{t-1}}\right)^{2k-2}. \tag{72}$$

Initially, for a small learning rate relative to the parameter magnitude ($\eta \ll x_0$), we have $\rho_t \approx 1 > \beta_2$, keeping the system in the **Coupled Phase**. However, as $x_t$ gets smaller, $\rho_t$ decreases monotonically. Crucially, once the threshold is crossed such that $\rho_t < \beta_2$, the system undergoes a phase transition. $v_t$ enters the **free-decay regime**, scaling as $v_t \propto \beta_2^t$. According to Lem. 5.5, this geometric decay of the denominator acts as an implicit exponential learning rate schedule, driving the parameter convergence at a linear rate determined by $\beta_2$:

$$|x_t| \propto \beta_2^{\frac{t}{2(k-2)}} \implies |g_t| = |x_t|^{k-1} \propto \beta_2^{\frac{t(k-1)}{2(k-2)}}. \tag{73}$$

This explains why the parametric trajectory is initially attracted to the non-trivial fixed point corresponding to linear convergence, even though that fixed point is ultimately unstable.

**Condition for Coupling of $m_t$ and $g_t$.** For this exponential convergence to be sustainable, a crucial asymmetry must be maintained: while $v_t$ must decouple to provide acceleration, the first moment $m_t$ must **remain coupled** to the gradient. This implies that $m_t$ must accurately track the direction and magnitude of the rapidly shrinking gradient $g_t$ rather than being dominated by stale historical momentum. Mathematically, this requires the decay factor of the gradient, denoted as $r_t := g_t/g_{t-1}$, to be *slower* (i.e., numerically larger) than the decay factor of the momentum history ($\beta_1$). The persistence condition is thus $r_t > \beta_1$.

Substituting the asymptotic decay rate of the gradient derived from the $v_t$-driven exponential stage:

$$r_t \approx \frac{g_t}{g_{t-1}} \approx \beta_2^{\frac{k-1}{2(k-2)}}. \tag{74}$$

Consequently, the boundary ensuring a persistent exponential convergence phase is given by:

$$\beta_1 < \beta_2^{\frac{k-1}{2(k-2)}}. \tag{75}$$

If this condition is violated ($\beta_1 > r_t$), $m_t$ decays slower than $g_t$, causing the momentum term to dominate the update excessively. This prevents the formation of a stable exponential trajectory, pushing the system into Regime III (no exponential stage).

However, sustained convergence necessitates local stability. Thus, while the condition in Eq. (75) induces a transient phase of linear convergence, the inherent instability of the fixed point eventually precipitates a spike (Regime II). True convergence is guaranteed only under the stricter condition $\beta_1 < \beta_2^{\frac{k}{2(k-2)}}$ (Regime I).

# F. Quadratic Case Analysis

In this section, we provide the fixed point analysis of quadratic case: $L(x) = \frac{1}{2}x^2$.

## F.1. Fixed Point Analysis

Similarly, consider Adam:

$$m_t = \beta_1 m_{t-1} + (1 - \beta_1)x_t^{k-1}$$
$$v_t = \beta_2 v_{t-1} + (1 - \beta_2)x_t^{2k-2}$$
$$x_{t+1} = x_t - \eta \cdot \frac{m_t}{\sqrt{v_t}}$$

Then we have:

$$x_{t+1} = \left(1 - \eta\frac{m_t}{x_t\sqrt{v_t}}\right)x_t = \left(1 - \eta\frac{m_t}{x_t^{k-1}} \times \frac{x_t^{k-2}}{\sqrt{v_t}}\right)x_t$$

For $k = 2$, define

$$\omega_t = \frac{m_t}{x_t^{k-1}}, \quad \lambda_t = \frac{1}{\sqrt{v_t}}.$$

Similar to Eq. (5), for $k = 2$, the iterative equation for the state variables is:

$$\begin{cases} \omega_{t+1} = \dfrac{\beta_1 \omega_t}{1 - \eta \omega_t \lambda_t} + 1 - \beta_1 \\[3mm] \lambda_{t+1} = \dfrac{\lambda_t}{\sqrt{\beta_2 + (1 - \beta_2)(\lambda_t x_{t+1})^2}} \\[3mm] x_{t+1} = (1 - \eta \omega_t \lambda_t) x_t \end{cases} \tag{76}$$

## 1. Calculation of Fixed Points.

Let the fixed point be denoted by $(\omega^*, \lambda^*, x^*)$. At a fixed point, the variables must satisfy $(z_{t+1} = z_t)$, yielding the following system:

$$\omega^* = \frac{\beta_1 \omega^*}{1 - \eta \omega^* \lambda^*} + 1 - \beta_1 \tag{77}$$

$$\lambda^* = \frac{\lambda^*}{\sqrt{\beta_2 + (1 - \beta_2)(\lambda^* x^*)^2}} \tag{78}$$

$$x^* = (1 - \eta \omega^* \lambda^*) x^* \tag{79}$$

Step 1: Analyze Eq. (79)

$$x^*(1 - (1 - \eta \omega^* \lambda^*)) = 0 \implies x^*(\eta \omega^* \lambda^*) = 0$$

Since we assume the learning rate $\eta > 0$, we have two cases: $x^* = 0$. $x^* \neq 0$, which implies $\omega^* \lambda^* = 0$.

Step 2: Analyze Eq. (78)

$$\lambda^* \sqrt{\beta_2 + (1 - \beta_2)(\lambda^* x^*)^2} = \lambda^*$$

This implies either $\lambda^* = 0$ or $\sqrt{\beta_2 + (1 - \beta_2)(\lambda^* x^*)^2} = 1$.

Assuming standard hyperparameters where $\beta_2 \in [0, 1)$: If $\lambda^* \neq 0$, then $\beta_2 + (1 - \beta_2)(\lambda^* x^*)^2 = 1 \implies (1 - \beta_2)(\lambda^* x^*)^2 = 1 - \beta_2 \implies (\lambda^* x^*)^2 = 1$. If $\lambda^* = 0$, the equation holds trivially (denominator is $\sqrt{\beta_2} \neq 0$).

Step 3: Analyze Eq. (77)

$$\omega^* \left(1 - \frac{\beta_1}{1 - \eta \omega^* \lambda^*}\right) = 1 - \beta_1$$

Combining constraints:

**Case A** ($x^* = 0$)**:** From Eq. (78), if $x^* = 0$, then $\lambda^* = \lambda^* / \sqrt{\beta_2}$. Since $\beta_2 < 1$, this requires $\lambda^* = 0$. Substituting $\lambda^* = 0$ into Eq. (77):

$$\omega^* = \beta_1 \omega^* + 1 - \beta_1 \implies \omega^*(1 - \beta_1) = 1 - \beta_1 \implies \omega^* = 1$$

This yields the fixed point $P_0 = (1, 0, 0)$.

**Case B** ($x^* \neq 0$)**:** From Step 1, we must have $\omega^* \lambda^* = 0$. If $\lambda^* = 0$: Eq. (77) gives $\omega^* = 1$. Eq. (78) is satisfied. If $\omega^* = 0$: Eq. (77) gives $0 = 0 + 1 - \beta_1 \implies \beta_1 = 1$, which contradicts $\beta_1 < 1$.

This results in a line of fixed points $(1, 0, c)$ for any $c \in \mathbb{R}$.

## 2. Local Stability Analysis

By calculation, the Jacobian matrix at the fixed point $\omega^* = 1, \lambda^* = 0, x^* \in \mathbb{R}$ is

$$M = \begin{pmatrix} 1 & \eta\beta_1 & 0 \\ 0 & \frac{1}{\sqrt{\beta_2}} & 0 \\ 0 & -\eta x^* & 1 \end{pmatrix}.$$

The eigenvalues of $M$ are $1, 1, \frac{1}{\sqrt{\beta_2}}$. Since $\frac{1}{\sqrt{\beta_2}} > 1$, the fixed point $\omega^* = 1, \lambda^* = 0, x^*$ is unstable for each $x^* \in \mathbb{R}$.

Therefore, for a quadratic loss, the Adam system possesses an unstable, trivial fixed point. Since the corresponding exponentially convergent fixed point does not exist, **the system cannot converge and dose not have stable convergence Regime I.**

### F.2. Convergence Rate Estimation Before Loss Spikes

For RMSProp $m_t = g_t$, if we give the learning rate an exponential scaling mechanism, that is, $\eta = \beta_2^{-t/2}$, we will see that it achieves initial superlinear convergence on quadratic functions. However, due to the stability constraints imposed by discreteness, a loss spike will eventually occur. The stability analysis for $k = 2$ is:

$$\begin{aligned} x_{t+1} &= x_t - \eta_t x_t = (1 - \eta_t \cdot 1)x_t, \\ \eta_{t+1} &= \gamma\eta_t, \quad (\gamma > 1). \end{aligned} \tag{80}$$

Thus this stability is necessarily unsatisfied and a spike necessarily occurs. The following continuous-time lemma describes the rate of superlinear convergence before the spike.

**Lemma F.1 (Super-Linear Convergence via Exponential Learning Rate Schedule for $k = 2$).** *Under the setup of the strongly convex objective $L(x) = \frac{1}{k}x^k$ ($k = 2$), consider the gradient flow dynamics with a time-varying learning rate $\eta(t)$:*

$$\frac{dx}{dt} = -\eta(t)\nabla L(x) = -\eta(t)x. \tag{81}$$

*If the learning rate follows an exponential schedule $\eta(t) = \eta_0 e^{\alpha t}$ with $\alpha > 0$ and $\eta_0 > 0$, then the solution $x(t)$ exhibits asymptotic super-exponential (double exponential) convergence:*

$$x(t) = x_0 \exp\left(\frac{\eta_0}{\alpha}\right) \exp\left(-\frac{\eta_0}{\alpha}e^{\alpha t}\right) \quad \text{as } t \to \infty. \tag{82}$$

*Proof.* We employ the separation of variables method. Rewriting the differential equation:

$$\frac{dx}{x} = -\eta_0 e^{\alpha t}dt. \tag{83}$$

Integrating both sides from the initial state $x_0$ to $x(t)$ (assuming $x_0, x(t) > 0$ for simplicity, without loss of generality):

$$\int_{x_0}^{x(t)} \frac{1}{y}dy = -\eta_0 \int_0^t e^{\alpha s}ds. \tag{84}$$

Computing the definite integrals. Note that for $k = 2$, the integral of $1/y$ results in the natural logarithm, distinct from the power law form for $k > 2$:

$$[\ln y]_{x_0}^{x(t)} = -\frac{\eta_0}{\alpha}(e^{\alpha t} - 1). \tag{85}$$

This simplifies to:

$$\ln x(t) - \ln x_0 = -\frac{\eta_0}{\alpha}(e^{\alpha t} - 1). \tag{86}$$

Rearranging terms to solve for $\ln x(t)$:

$$\ln x(t) = \ln x_0 + \frac{\eta_0}{\alpha} - \frac{\eta_0}{\alpha}e^{\alpha t}. \tag{87}$$

Exponentiating both sides to retrieve $x(t)$:

$$x(t) = \exp\left(\ln x_0 + \frac{\eta_0}{\alpha}\right) \exp\left(-\frac{\eta_0}{\alpha}e^{\alpha t}\right). \tag{88}$$

Defining the constant coefficient $C = x_0 e^{\eta_0/\alpha}$, we obtain:

$$x(t) = C \exp\left(-\frac{\eta_0}{\alpha} e^{\alpha t}\right). \tag{89}$$

As $t \to \infty$, the term inside the exponent grows exponentially, driving $x(t)$ to zero at a super-exponential rate, significantly faster than the linear convergence observed in the constant learning rate case or the $k \geq 4$ case. □

However, with the introduction of the Adam momentum mechanism, **if $g_t$ decays too quickly (exceeding the exponential rate $\beta_1$), $m_t$ will be bottlenecked by its own memory**, so the initial convergence before spike of the quadratic function is still linear.

**Lemma F.2** (**Momentum-Limited Convergence under Exponential Schedule for** $k = 2$). *Consider the quadratic objective $L(x) = \frac{1}{2}x^2$ optimized by continuous-time gradient descent with momentum. The dynamics are governed by the system:*

$$\dot{m} = -(1 - \beta_1)m + (1 - \beta_1)x, \tag{90}$$

$$\dot{x} = -\eta(t)m, \tag{91}$$

*where $1 - \beta_1 > 0$ represents the friction coefficient. Assume $1 - \beta_1 > \alpha$. If the learning rate grows exponentially as $\eta(t) = \eta_0 e^{\alpha t}$ (with $\alpha > 0$), the parameter $x(t)$ does **not** achieve super-exponential convergence. Instead, the convergence is asymptotically bounded by an exponential envelope determined by the momentum decay and the schedule growth rate:*

$$|x(t)| \lesssim C \exp\left(-\frac{(1 - \beta_1) - \alpha/2}{2}t\right). \tag{92}$$

*This indicates that the "memory" of the momentum term acts as a bottleneck, restricting the system to linear convergence (exponential in time) despite the exponentially growing learning rate.*

*Proof.* We reduce the system of two first-order ODEs to a single second-order ODE governing $x(t)$. From Eq. (91), we have $m = -\frac{\dot{x}}{\eta(t)}$. Differentiating this with respect to time yields:

$$\dot{m} = -\frac{\ddot{x}\eta(t) - \dot{x}\dot{\eta}(t)}{\eta(t)^2} = -\frac{\ddot{x}}{\eta(t)} + \frac{\dot{\eta}(t)}{\eta(t)}\frac{\dot{x}}{\eta(t)}. \tag{93}$$

Let $\lambda = 1 - \beta_1$. Substituting expressions for $m$ and $\dot{m}$ into Eq. (90):

$$-\frac{\ddot{x}}{\eta} + \frac{\dot{\eta}}{\eta}\frac{\dot{x}}{\eta} = -\lambda\left(-\frac{\dot{x}}{\eta}\right) + \lambda x. \tag{94}$$

Multiplying through by $-\eta(t)$ and rearranging terms:

$$\ddot{x} + \left(\lambda - \frac{\dot{\eta}}{\eta}\right)\dot{x} + \lambda\eta(t)x = 0. \tag{95}$$

Substituting the schedule $\eta(t) = \eta_0 e^{\alpha t}$, we have $\frac{\dot{\eta}}{\eta} = \alpha$. The dynamics simplify to a damped harmonic oscillator with time-varying stiffness:

$$\ddot{x} + (\lambda - \alpha)\dot{x} + \omega^2(t)x = 0, \tag{96}$$

where the effective stiffness is $\omega^2(t) = \lambda\eta_0 e^{\alpha t}$. This equation describes an oscillator where the restoring force grows exponentially, but the damping coefficient $(\lambda - \alpha)$ is constant. For stability, we assume $\lambda > \alpha$ (momentum decay is faster than LR growth). To analyze the asymptotic behavior as $t \to \infty$, we employ the Liouville-Green (WKB) approximation. For an equation of the form $\ddot{x} + \gamma\dot{x} + \omega^2(t)x = 0$ with slowly varying or monotonic $\omega(t)$, the asymptotic solution is given by:

$$x(t) \approx \frac{C}{\sqrt{\omega(t)}} \exp\left(-\frac{\gamma}{2}t\right) \cos\left(\int \omega(s)ds + \phi\right). \tag{97}$$

In our case, $\gamma = \lambda - \alpha$ and $\omega(t) \propto e^{\alpha t/2}$. Thus, the amplitude envelope $A(t)$ evolves as:

$$A(t) \propto (e^{\alpha t/2})^{-1/2} \cdot \exp\left(-\frac{\lambda - \alpha}{2}t\right) \tag{98}$$

$$= e^{-\alpha t/4} \cdot e^{-\frac{\lambda}{2}t + \frac{\alpha}{2}t} \tag{99}$$

$$= \exp\left(-\frac{\lambda - \frac{\alpha}{2}}{2}t\right). \tag{100}$$

Substituting $\lambda = 1 - \beta_1$, the decay rate is governed by the exponent $\frac{(1-\beta_1)-\alpha/2}{2}$. Crucially, this expression is linear in $t$, implying standard exponential convergence. The momentum term prevents the trajectory from following the super-exponential acceleration that would occur in the memoryless setting. $\qquad\square$

### F.3. Empirical Verification of Convergence Rates on Quadratic Loss

Figure 9 illustrates the training dynamics of Adam and RMSProp on the simple quadratic objective $L(x) = \frac{1}{2}x^2$. As shown in Figure 9(a), RMSProp exhibits superlinear convergence prior to the occurrence of the loss spike. In contrast, Adam maintains a stable linear convergence, with a decay rate that aligns precisely with the theoretical slope of $\frac{(1-\beta_1)-\alpha/2}{2}$, where $\alpha = -\frac{1}{2}\ln(\beta_2)$.

Figures 9(b) and (c) analyze the relationship between the second moment estimate $v_t$ and the squared gradient $g_t^2$ during the training of RMSProp and Adam, respectively. It is evident that $g_t^2$ decays significantly faster than $v_t$. Consequently, $v_t$ decouples from the gradient and enters a regime of free decay, evolving as $v_t \approx \beta_2 v_{t-1}$.

Figure 9(d) depicts the evolution of the first moment (momentum) and the raw gradient during Adam training. Unlike the second moment, the first moment remains tightly coupled with the gradient throughout the optimization process.

Below, we present the experimental phase diagram of Adam hyperparameters for quadratic loss.

**Experiment settings** The initialization is set as $x_0 = 1.005$ with a learning rate of $\eta = 0.01$, while $\beta_1$ and $\beta_2$ are sampled from a uniform $50 \times 50$ grid spanning the region $[0.01, 0.99]^2$. Using this initialization and learning rate is to avoid the case where signGD ($\beta_1 = \beta_2 = 0$) identifies the exact minimum.

Observed column-wise, the minimum loss increases as $\beta_2$ decreases for a fixed $\beta_1$. This phenomenon aligns with the findings in (Bai et al., 2025): a larger $\beta_2$ induces a lagged response of $v_t$ to $g_t^2$. Specifically, while $g_t$ may decrease to a smaller value, $v_t$ remains relatively large due to this delay. This sustained magnitude ensures that the preconditioner does not exceed its threshold, thereby allowing the model to reach a lower loss.

Observed row-wise, with $\beta_2$ held constant, the minimum loss gradually increases as $\beta_1$ increases. A larger $\beta_1$ elevates the preconditioning threshold. Assuming the initial scale and decay rate of $v_t$ are identical (given a fixed $\beta_2$), a larger $\beta_1$ results in a delayed spike. However, a larger $\beta_1$ also increases the scale of overshooting and causes it to occur earlier in the optimization phase. This is evidenced by the earlier appearance of small-scale oscillations in the loss curve. In this scenario, the momentum causes overshooting before the loss can reach lower levels.

## G. Supplementary Experiments

In this section, we present additional experimental results that complement the findings in the main text.

### G.1. Zoom-in Analysis of the Phase Diagram

The stability conditions derived in Theorem 4.1 define the boundaries of the convergence region. Figure 11 provides the theoretical and empirical phase diagrams for the case $k = 6$, showing consistent alignment between theory and experiment.

For the case $k = 4$, the theoretical stability region (where the spectral radius $\rho(J) < 1$) is primarily governed by the inequality $\beta_1 < \beta_2^{\frac{k}{2(k-2)}}$, which simplifies to $\beta_1 < \beta_2$. However, a secondary constraint appears in the lower-left corner (small $\beta_1$ and $\beta_2$), as shown in Figure 12(a) and magnified in Figure 12(b).

Figures 12(c) and (d) display the heatmaps of the final training loss for the main region and the zoomed-in lower-left region, respectively. In the main region, the theoretical boundary for local stability aligns precisely with the experimental

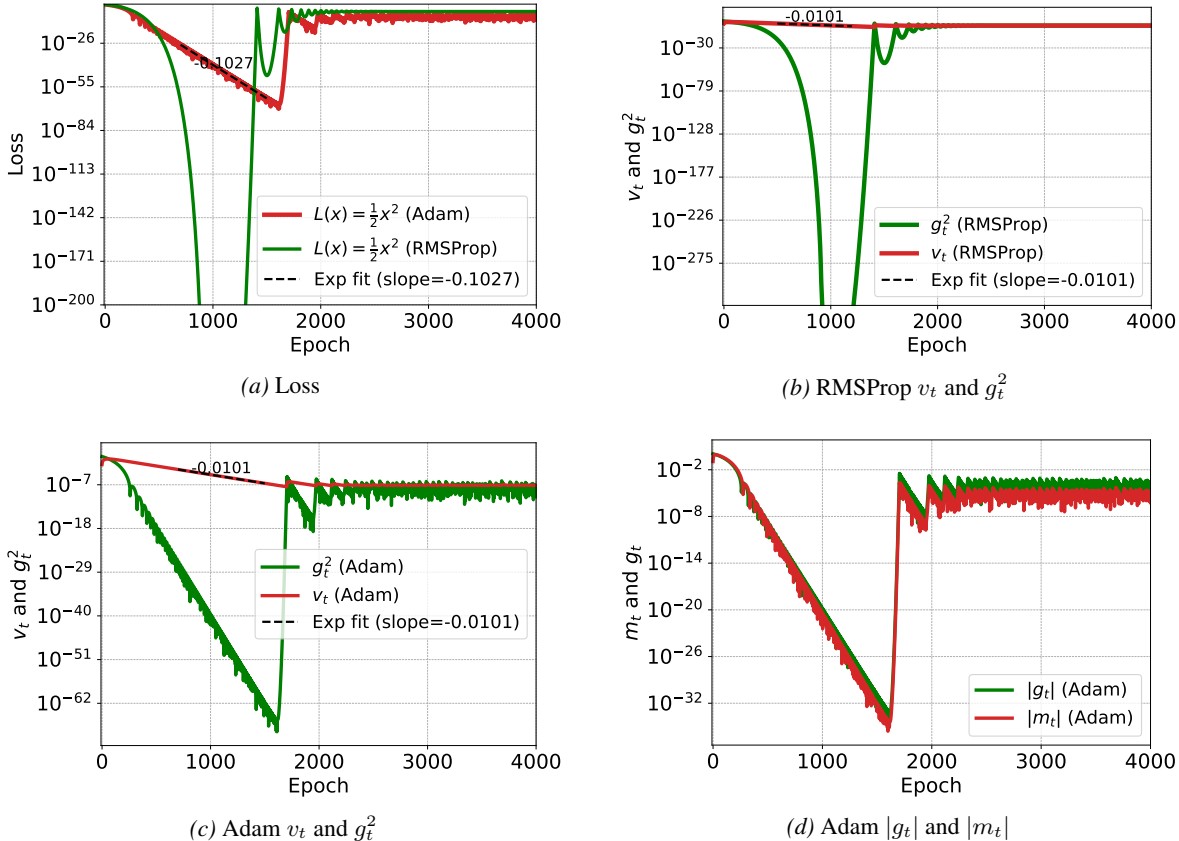

*Figure 9.* Training dynamics on quadratic loss $L(x) = \frac{1}{2}x^2$. (a) Loss curves for Adam and RMSProp. (b-c) Comparison of second moment $v_t$ and squared gradient $g_t^2$ for RMSProp and Adam. (d) Comparison of first moment $m_t$ and gradient $g_t$ for Adam.

convergence boundary. However, discrepancies emerge in the lower-left corner: the experiments indicate that the loss may still converge in regions theoretically classified as locally unstable.

Figure 13 elucidates the mechanism behind this discrepancy. We fix $\beta_1 = 0.001$ and vary $\beta_2$ to observe the transition in dynamics:

- **Stable Region:** When $\beta_2$ is sufficiently large, the loss exhibits smooth linear convergence (Figure 13(a)).

- **Period-Doubling:** As $\beta_2$ decreases, the fixed point becomes unstable, leading to period-doubling bifurcations and chaos. However, as long as the effective sharpness satisfies $u_t < 2/\eta$, the system remains bounded and converges, characterized by "sawtooth" fluctuations in the loss curve (Figure 13(b)).

- **Instability:** Further decreasing $\beta_2$ leads to different behaviors depending on the existence of non-trivial fixed points. The system may exhibit transient linear convergence followed by a spike or pure oscillation (Figure 13(c)).

Figures 13(d) and (e) plot the bifurcation diagrams of the effective sharpness $u_t$ over 500 steps. For low momentum ($\beta_1 = 0.001$), even after the fixed point loses stability, the system undergoes a period-doubling route to chaos while maintaining $u_t < 2/\eta$, allowing for convergence. In contrast, for high momentum ($\beta_1 = 0.9$), the onset of instability causes $u_t$ to immediately exceed the $2/\eta$ threshold, preventing convergence.

### G.2. Experiment settings

**Fig. 15(c)** A two-layer neural network trained on the MNIST dataset using Cross-Entropy Loss. The architecture employs different activation ReLU and Softmax. The learning rate is ranging from $[0.0005, 0.001, 0.005, 0.01, 0.05, 0.1, 0.5]$.

**Fig. 15(f)** The Transformer model is GPT-2 trained on a subset split from the SlimPajama dataset. The learning rate is

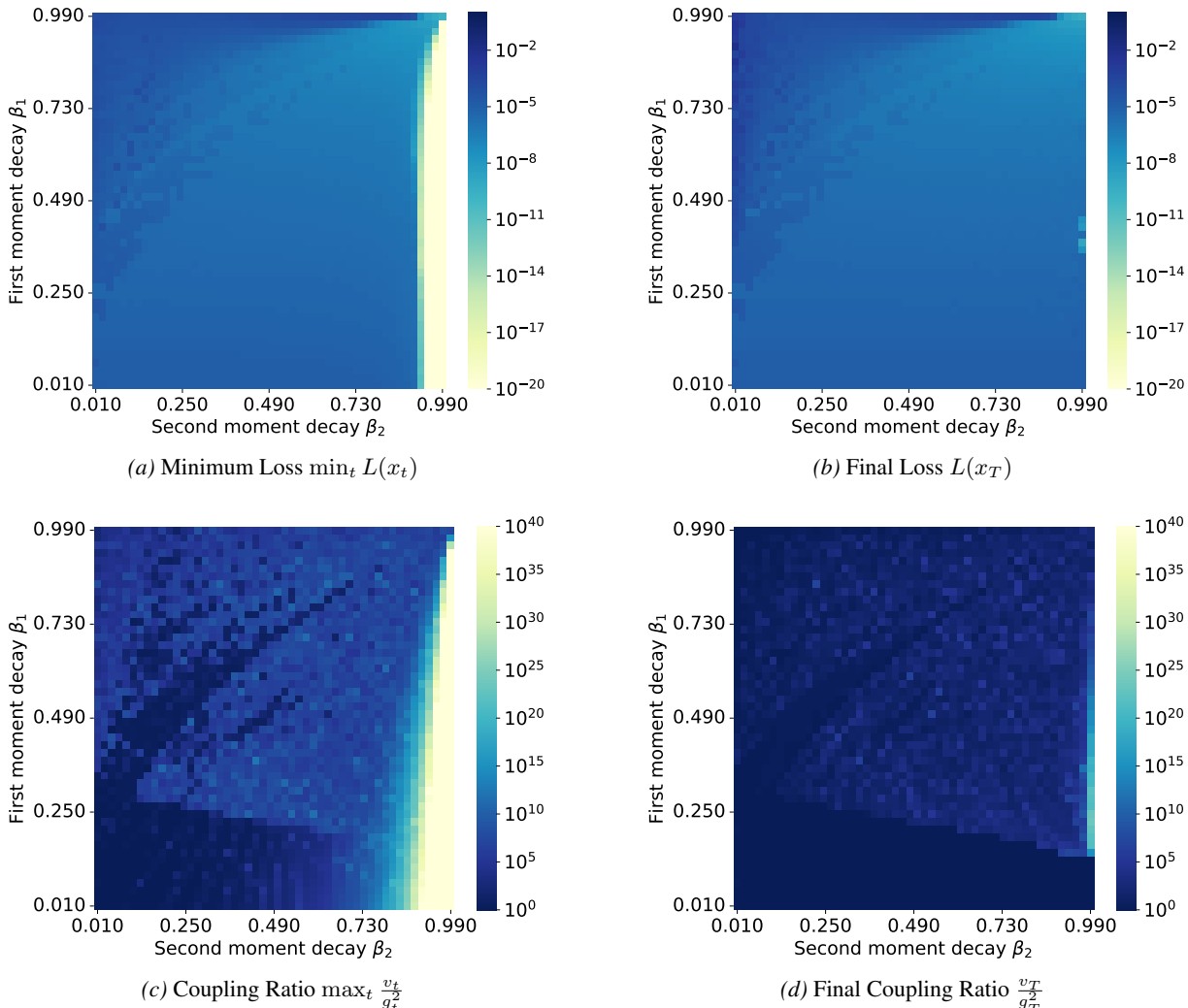

*(a)* Minimum Loss $\min_t L(x_t)$

*(b)* Final Loss $L(x_T)$

*(c)* Coupling Ratio $\max_t \frac{v_t}{g_t^2}$

*(d)* Final Coupling Ratio $\frac{v_T}{g_T^2}$

*Figure 10.* Empirical phase diagrams of Adam optimizing $f(x) = \frac{x^2}{2}$. A sliding average filter with a window size of 100 is applied to smooth numerical fluctuations. (a) and (b) show the minimum and final loss of the optimization process, respectively. (c) and (d) display the maximum and final values of the ratio $\frac{v_t}{g_t^2}$.

ranging from $[0.00005, 0.0001, 0.0005, 0.001, 0.005]$. The optimal learning rate is $0.001$ for both Adam and SGD. The CNN model comprises two convolutional layers and is trained on the CIFAR-10 dataset. The learning rate is ranging from $[0.0005, 0.001, 0.005, 0.01, 0.05, 0.1]$. The optimal learning rate is $0.005$ for Adam and $0.05$ for SGD.

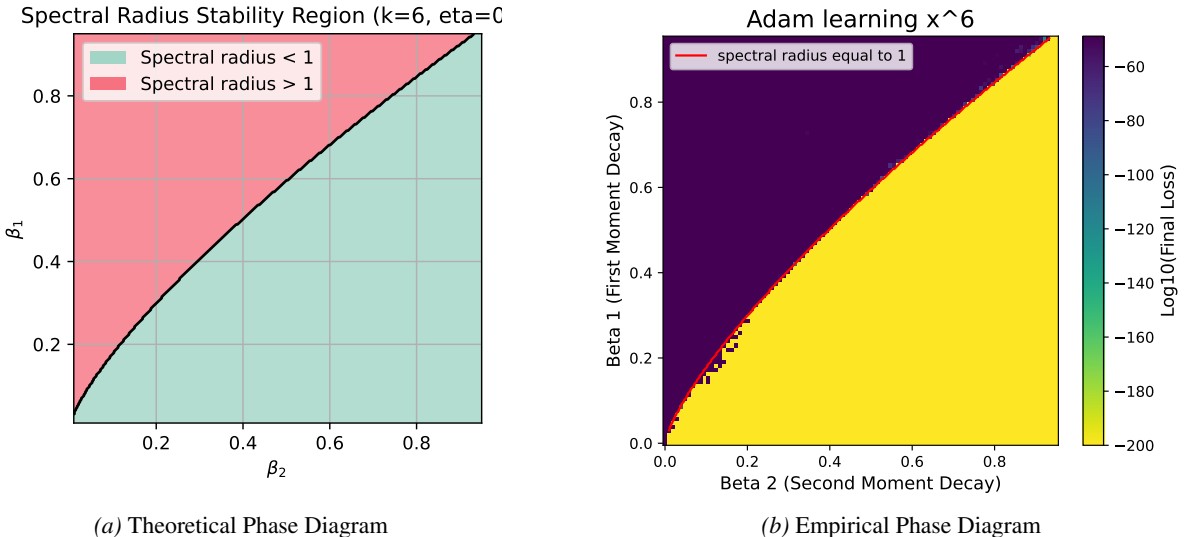

*(a)* Theoretical Phase Diagram          *(b)* Empirical Phase Diagram

*Figure 11.* **Phase diagrams for degree** $k = 6$**.** (a) Theoretical convergence region partitioned by the stability conditions in Eq. (8). (b) Experimental validation using Adam on $L(x) = \frac{1}{6}x^6$ ($x_0 = 1.0, \eta = 0.001$). The heatmap displays the final training loss after $100,000$ steps.

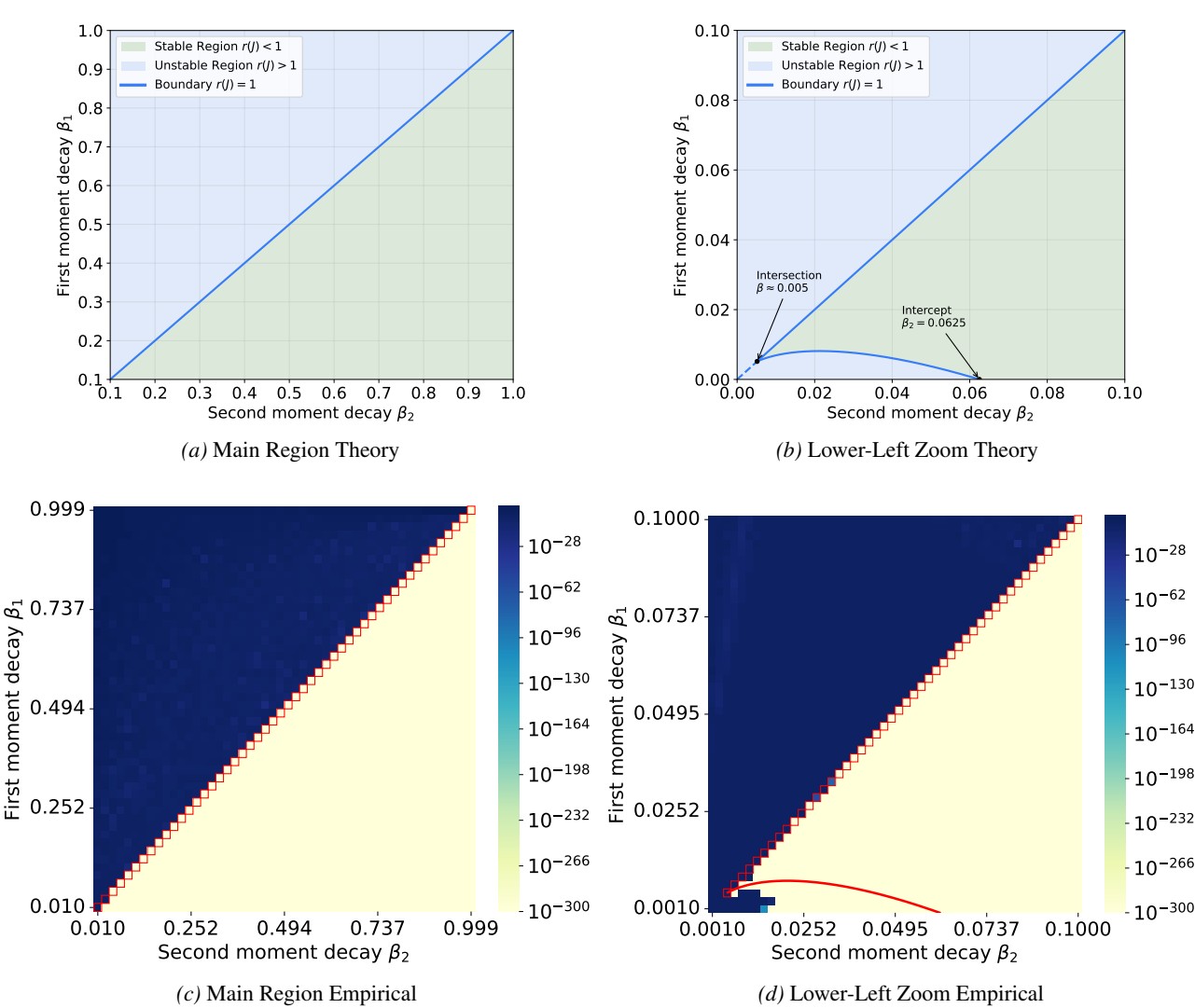

*Figure 12.* **Zoom-in analysis for** $k = 4$. (a) Theoretical stability boundary for the full parameter range. (b) Magnified theoretical boundary for the lower-left corner ($\beta_2 \in [0, 0.1]$). (c-d) Corresponding experimental loss heatmaps. Note the convergence observed in theoretically unstable areas in (d).

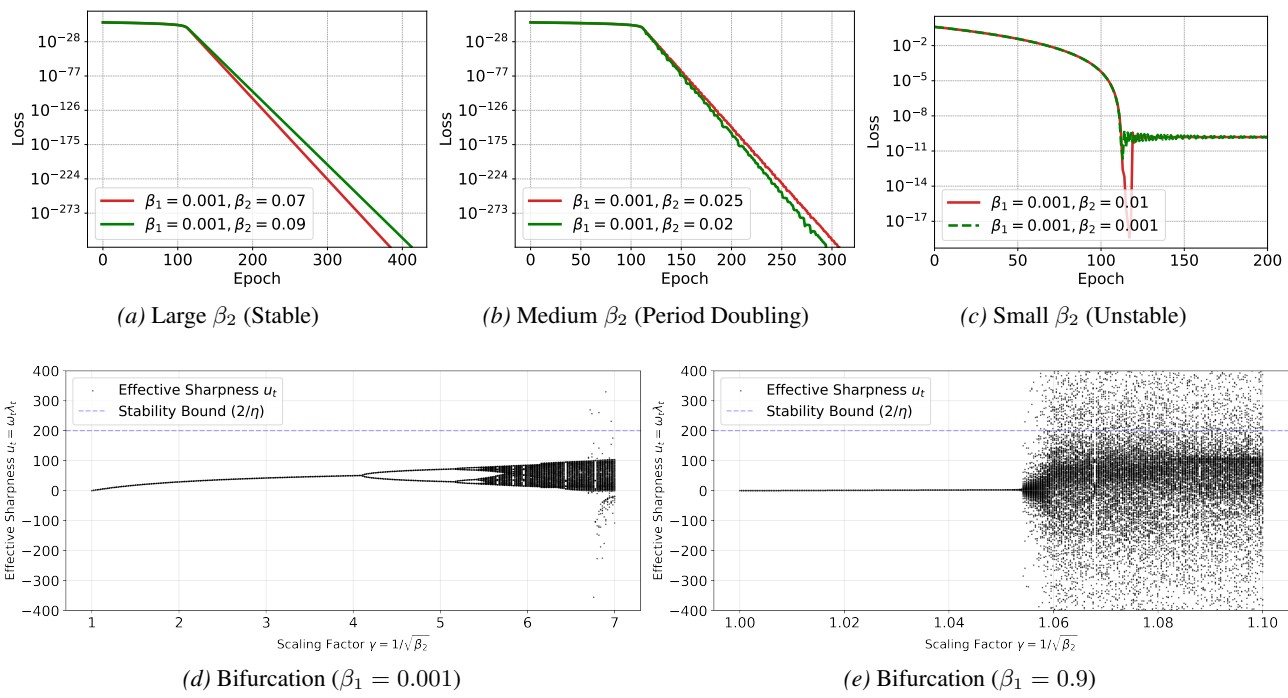

*Figure 13.* **Dynamics and Bifurcation Analysis.** (a-c) Loss trajectories for fixed $\beta_1 = 0.001$ with decreasing $\beta_2$. (d-e) Bifurcation diagrams of effective sharpness $u_t$ versus $\beta_2$. The blue dashed line indicates the stability threshold $2/\eta$. Low $\beta_1$ allows the system to remain bounded (below threshold) despite local instability, whereas high $\beta_1$ leads to immediate divergence.

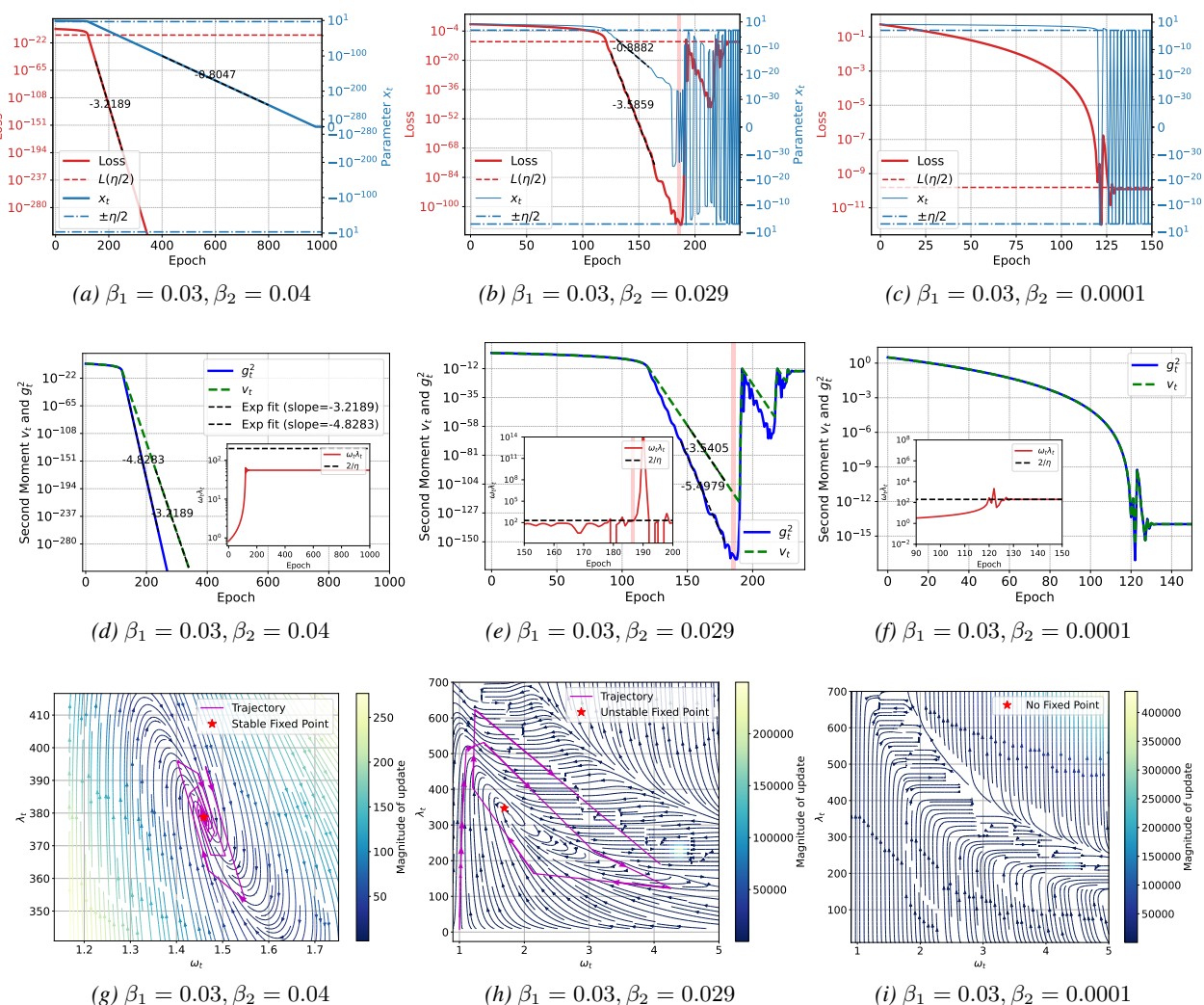

*Figure 14.* Three distinct dynamical regimes of Adam on the degenerate objective $L(x) = \frac{1}{4}x^4$ with learning rate $\eta = 0.01$. **Top Row:** Evolution of training loss (red, left axis) and parameter trajectory $x_t$ (blue, right axis). The text online shows the slope of the exponential fit. **Middle Row:** Evolution of the second moment estimate $v_t$ versus the squared gradient $g_t^2$. Insets display the evolution of the stability metric $\omega_t \lambda_t$ relative to the theoretical threshold $2/\eta$. **Bottom Row:** Vector fields of $\omega$ and $\lambda$; purple lines represent evolutionary trajectories. **(Left Column) Regime I: Stable Exponential Convergence.** $v_t$ completely decouples from $g_t^2$, facilitating stable exponential acceleration. **(Middle Column) Regime II: Exponential Convergence followed by Spike.** $v_t$ initially decouples, driving acceleration. However, the stability condition is persistently violated ($\omega_t \lambda_t > 2/\eta$) due to response lag, triggering a violent loss spike. **(Right Column) Regime III: SignGD-like Oscillation.** $v_t$ tracks $g_t^2$ closely (tight coupling), preventing the exponential convergence phase. Violations of the stability threshold ($\omega_t \lambda_t > 2/\eta$) trigger an instantaneous correction, resulting in oscillations.

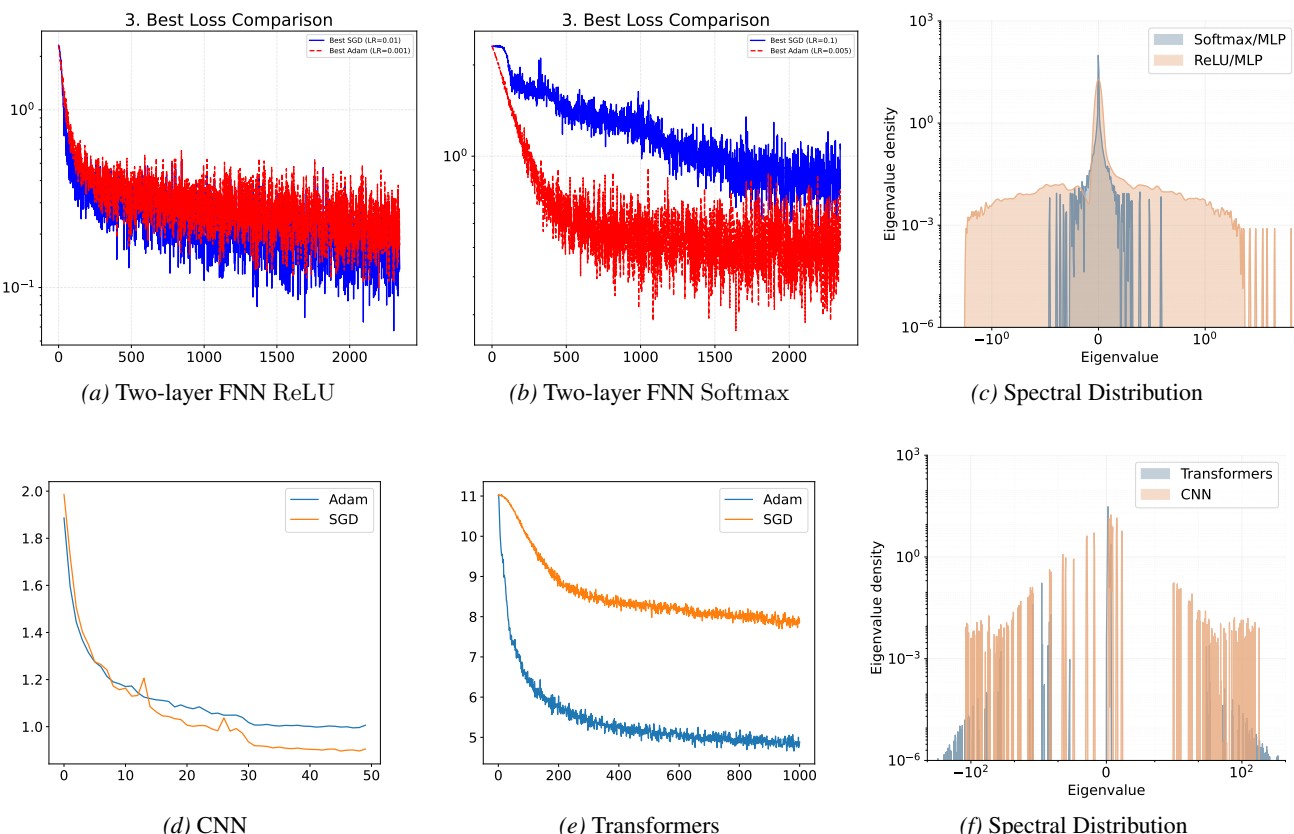

*Figure 15.* Evolution of training loss across different architectures and optimizers, using learning rates tuned for optimal performance. **(a, d)** Adam and SGD perform comparably on the two-layer FNN with ReLU and the CNN. **(b, e)** Adam outperforms SGD on the two-layer FNN with Softmax and Transformers. **(c, f)** Hessian spectral densities of different model architectures at initialization. The densities are smoothed using KDE. (c) A two-layer FNN on MNIST with Softmax (blue) and ReLU (orange) activations. (f) GPT-2 (blue) and a two-layer CNN on CIFAR-10 (orange).

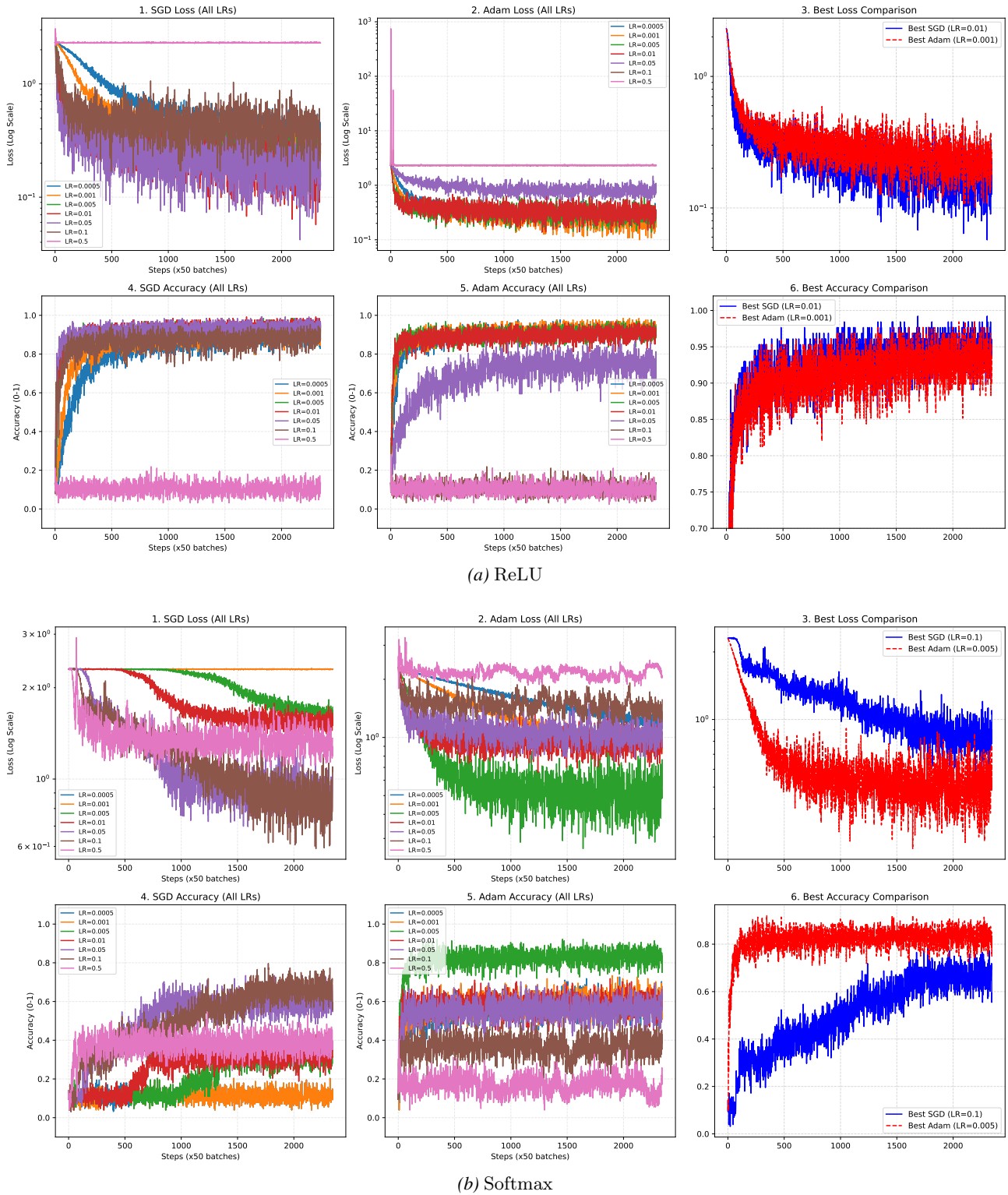

*(a)* ReLU

*(b)* Softmax

*Figure 16.* Raw loss curves of a two-layer FNN trained with Adam and SGD across various learning rates. (a) ReLU activation. (b) Softmax activation.

