# OpenReview forum: "Towards Understanding Adam Convergence on Highly Degenerate Polynomials"
_ICML.cc/2026/Conference — ICML 2026 spotlight_

### Official Review · Reviewer_gsC6 · 2026-03-02

**Soundness:** 3
**Presentation:** 3
**Significance:** 2
**Originality:** 3
**Overall Recommendation:** 4
**Confidence:** 4

**Summary:**

This paper investigated the specific class of objective functions where Adam exhibits inherent advantages. The authors identify the highly degenerate polynomials where Adam outperforms GD and Momentum. This phenomenon stems from a decoupling mechanism between the second moment $v_t$ and squared gradients $g_t^2$, which exponentially amplifies the effective learning rate. Furthermore, they characterize Adam’s hyperparameter phase diagram, identifying three distinct behavioral regimes: stableconvergence, spikes, and SignGD-like oscillation.

**Compliance With Llm Reviewing Policy:**

Affirmed.

**Final Justification:**

My concerns in the original review are fully addressed by the authors' rebuttal. I think this paper can be published in ICML.

**Key Questions For Authors:**

No further questions

**Limitations:**

yes

**Strengths And Weaknesses:**

**Strengths:**

(1) The problem investigated in this paper is very interesting and may potentially provide some new insights for the convergence of Adam.

(2) The phenomenon reported in this paper is new, especially the identification of three distinct behavioral regimes for Adam.

(3) The experiments are comprehensive and support their theoretical observations.

**Weaknesses:**

(1) The main weakness is the function class investigated in this paper, univariate polynomials, is too simple.

(2) This paper only investigated the deterministic setting. In practice, Adam is mainly implemented in stochastic setting (mini-batch setting), not deterministic setting (full-batch setting).

---

> ### Author Rebuttal · Authors · 2026-03-28
>
> We sincerely thank the reviewer for the positive assessment and thoughtful comments. Our responses are as follows.
>
> **W1: The main weakness is the function class investigated in this paper, univariate polynomials, is too simple.**
>
> We thank the reviewer for this comment. We try to explain this from three aspects:
>
> **(1) Theoretical Necessity of Isolation.** Near any local minimum, a Taylor expansion decomposes the loss into strongly convex and degenerate directions; our $L(x) = \frac{1}{k}x^k$ precisely captures the latter. This isolation allows us to rigorously establish the $v_t$–$g_t^2$ decoupling mechanism, derive tight stability conditions, and obtain an explicit convergence rate $\beta_2^{1/(2(k-2))}$ — all with strong empirical alignment.
>
> **(2) The Results Extend to a Broader Function Class.** Theorem 4.1 can be rigorously extended to any real analytic function with a degenerate local minimum of order $k \ge 4$ (i.e., general objectives that can be locally written as $L(x) = \frac{c}{k}x^k(1+h(x))$, where $c>0$, $h$ is analytic, and $h(0)=0$). Since $g(x) = cx^{k-1}(1+h(x) + \frac{1}{k}xh'(x)) = \Theta(x^{k-1})$ as $x\to 0$, the higher-order terms from $h$ vanish when evaluating the Jacobian at the fixed point $(\omega^\*, \lambda^\*, 0)$. The stability conditions (Eqs. 8–9) and convergence rate (Eq. 10) are therefore **identical** for this entire function class. Our results thus characterize the universal behavior of any loss with a $k$-th order degenerate minimum, not merely the specific monomial. We will incorporate the generalization as a formal remark in the revised manuscript.
>
> **(3) High-Dimensional Mixed-Curvature Analysis.** We have already analyzed the mixed-curvature case in Section 7 and quadratic theory in Appendix F. In landscapes coupling quadratic and degenerate directions, the optimization bottleneck is determined by the flattest directions — precisely those our model captures. Furthermore, Hessian spectral analysis of real networks (Fig. 15(c,f)) consistently shows eigenvalues concentrated near zero, directly supporting the prevalence of degenerate directions in practice. We agree this does not constitute a formal proof for the general neural network case, and we will add a explicit discussion of in the limitation section.
>
> **W2: This paper only investigates the deterministic setting. In practice, Adam is mainly used in the stochastic (mini-batch) setting.**
>
> We respond from two angles:
>
> **(1) The Deterministic Setting Is Well-Motivated.** Although Adam's advantage under heavy-tailed stochastic noise was proposed early on Zhang et al., (2020), a key subsequent finding by Kunstner et al. (2023) demonstrated that the performance gap between Adam and GD persists even in deterministic full-batch settings. This observation motivated a line of work specifically studying Adam's advantage under deterministic conditions — the setting our paper also adopts.
>
> **(2) Our Mechanism Is Complementary to Stochastic Explanations.** Prior mechanisms each rely on specific structural assumptions. The table below situates our work among prior analyses:
>
> | Work | Proposed Mechanism | Key Assumption | Requires Stochasticity |
> |---|---|---|---|
> | Zhang et al. (2020) | Heavy-tailed noise | Heavy-tailed gradient noise | Yes |
> | Zhang et al. (2024) | Hessian block heterogeneity | Block-diagonal Hessian structure | No |
> | Tomihari & Sato (2025) | Gradient heterogeneity | Heterogeneous gradients and SignGD approximation | No |
> | Xie et al. (2024) | $\ell_\infty$ geometry exploitation | Coordinate-wisely smooth w.r.t. $\ell_\infty$ norm | No |
> | Kunstner et al. (2024) | Heavy-tailed class imbalance | Imbalanced data distribution and SignGD approximation| No |
> | **Ours** | **$v_t$–$g_t^2$ decoupling on degenerate directions** | **Loss landscape degeneracy** | **No** |
>
> Our mechanism requires only that degenerate directions exist — a structural property that holds universally in deep networks, independent of data distribution, Hessian block structure, or noise characteristics. These mechanisms are not mutually exclusive; multiple factors likely operate simultaneously in real training, and our work **isolates one that has been overlooked**. We will clarify this in the limitation section.
>
> We thank you again for helping us improve the quality of our manuscript. Should any further questions remain, we are fully committed to providing additional clarification and discussion.

---

> > ### Author Rebuttal · Reviewer_gsC6 · 2026-04-01
> >
> > Thank you for the authors' response. My concerns are almost addressed by the rebuttal. I will maintain the **Weak Accept** evaluation. I hold no strong preference for or against this paper. I am OK with whatever decision the Area Chair makes regarding its acceptance or rejection.

---

> > > ### Author Response · Authors · 2026-04-08
> > >
> > > We sincerely appreciate the reviewer's positive score and constructive suggestions, as well as their continued engagement with our manuscript.

---

### Official Review · Reviewer_EWWF · 2026-03-09

**Soundness:** 3
**Presentation:** 3
**Significance:** 3
**Originality:** 2
**Overall Recommendation:** 3
**Confidence:** 4

**Summary:**

This paper provides a comprehensive convergence analysis of Adam and Gradient Descent on the highly degenerated polynomial function, $y={1\over k}x^{k}, k \ge 4$. The theoretical results indicate that Adam and RMSProp can achieve the linear convergence rate while continuous GD can only achieve the sub-linear convergence. These results also provide the necessary conditions for the hyper-parameters. The authors also provide some empirical results to verify their theoretical findings.

**Compliance With Llm Reviewing Policy:**

Affirmed.

**Key Questions For Authors:**

Is it possible to extend the objective function scope to some more general ones? Such as considering $L(x) = {1 \over k}x^k + b$ or other highly degenerate functions that may be out of scope of polynomials.

**Strengths And Weaknesses:**

**Strengths**

(a). In my view, this paper provides some interesting results for showing why Adam could outperforme GD/SGD in the special loss function class: highly degenerated polynomial function. Overall, the theoretical results look sound and novel.

(b). The requirement for hyperparameters are more delicated than the commonly known $\beta_1 < \beta_2$ for Adam.

(c). The experiments are also comprehensive.

**Weaknesses**

(a). An important motivation of this paper is that the highly degenerated polynomial function could commonly appear in the deep learning, as the authors indicate in the Introduction part. However, it's better to provide more evidence for illustrating this point, such as some specific examples. Also, there is lack of comprehensive comparison with exisiting results.

(b). The objective function is one a one-dimensional form which can be relatively simple. Therefore, the whole proof may not contain enough technical novelty.

(c). The theoretical results for GD only involves the continuous version and misses the discrete version while the results for Adam are based on its discrete version.

(d). Though the results are new, considering the relatively simple form of objective functions, the proof may be not so technical.

---

> ### Author Rebuttal · Authors · 2026-03-28
>
> We sincerely thank the reviewer for high appreciation of the importance of our work. Below, we provide point-by-point clarifications addressing your key concerns.
>
> **W1-1: Lack of evidence that degenerate polynomials commonly appear in deep learning.**
>
> We address this concern from two angles:
>
> **(1) Degenerate Polynomials as a Universal Local Model.** Taylor expansion near any local minimum is standard practice. Decomposing the local landscape into strongly convex directions and degenerate directions is a natural and well-motivated decoupling. Our prototype $L(x) = \frac{1}{k}x^k$ precisely captures the degenerate component of this decomposition — it is not an artificial construction but the canonical local model for any minimum where the first $k-1$ derivatives vanish. The isolation allows us to rigorously establish the $v_t$–$g_t^2$ decoupling mechanism and derive tight, falsifiable stability conditions, without the analysis being obscured by high-dimensional interactions.
>
> **(2) High-Order Degeneracy is a Structural Feature of Deep Networks.** Far from being a toy assumption, high-order degeneracy is a well-documented and theoretically grounded property of deep learning loss landscapes. We summarize the key evidence below:
> | Source | Finding | Implication |
> |---|---|---|
> | Sagun et al. (2017) | Empirical Hessian eigenvalues of overparameterized networks are heavily concentrated near zero | Degenerate directions are pervasive in practice |
> | Fukumizu et al. (2019); Simsek et al. (2021) | Network parameterization symmetries create flat directions in the loss landscape | Degeneracy is structurally induced by overparameterization |
> | Zhang et al. (2021; 2022b); Bai et al. (2024) | Critical points of smaller networks embed as high-dimensional degenerate manifolds into larger networks (Embedding Principle) | Degeneracy grows with depth and width |
> | Zhang et al. (2025) | In two-layer networks, the target set with zero generalization error contains strictly degenerate convergence directions | Degenerate directions exist even at globally optimal solutions |
>
> A concrete example illustrates the mechanism behind the Embedding Principle: for the shallow linear network $f_\theta = W_2 W_1$, the origin is a non-degenerate saddle point. Deepening to $f_\theta = W_3 W_2 W_1$ renders the origin degenerate — the loss along certain directions vanishes to higher order. High-order degeneracy is therefore not an artificial toy case but a fundamental topological consequence of overparameterization and depth.
>
> **W1-2: There is a lack of comprehensive comparison with existing results.**
>
> Due to space limitations, we refer the reviewer to the comparison table in our response to Reviewer x93G. We will incorporate this comparison more clearly in the revised manuscript.
>
> **Q1: Extension to more general function classes?**
>
> **The results indeed can extend to a broader function class.** Theorem 4.1 extends directly to $L(x) = \frac{1}{k}x^k(1+h(x)) (k\geq 4)$ where $h$ is analytic and $h(0)=0$. Since $g(x) = x^{k-1}(1+h(x) + \frac{1}{k}xh'(x)) = \Theta(x^{k-1})$ as $x\to 0$, the higher-order terms from $h$ vanish when evaluating the Jacobian at the fixed point $(\omega^\*, \lambda^\*, 0)$. The stability conditions (Eqs. 8–9) and convergence rate (Eq. 10) are therefore **identical** for this entire function class. Our results thus characterize the universal behavior of any loss with a $k$-th order degenerate minimum, not merely the specific monomial. We will incorporate the generalization as a formal remark in the revised manuscript.
>
> **W2 & W4: Technical Novelty of the 1D Objective.**
>
> We respectfully note that technical complexity alone should not define theoretical contribution — what matters is whether it delivers genuine insight.
>
> (1) The core challenge lies not in dimension but in analyzing the nonlinear fixed-point stability of a three-variable coupled system $(\omega_t, \lambda_t, x_t)$.
>
> (2) The decoupling mechanism itself where $v_t$ detaches from $g_t^2$ is a genuine conceptual discovery.
>
> (3) The proof of global convergence (Thm. 5.7) relies on Coppel’s (1955) fixed-point theory, which is not standard and trivial.
>
> **W3: GD convergence result is only for continuous-time, while Adam result is discrete.**
>
> **(1) Continuous vs. discrete does not affect convergence order.** Continuous-time GD is the limit of discrete GD; if gradient flow is sublinear, discrete GD has the same order (up to constants).
>
> **(2) Discreteness is essential for stability and Adam’s behavior.** For GD, gradient flow always converges while discrete GD's stability condition is well-known: $\eta < 2/\lambda_{\max}$. For Adam, however, discreteness is crucial: stability depends nontrivially on hyperparameters, and continuous models miss key phenomena such as loss spikes. These arise from discrete stability constraints and underlie the phase diagram. Hence, preserving discreteness is necessary to capture Adam’s true dynamics, rather than an asymmetry.

---

> > ### Author Rebuttal · Reviewer_EWWF · 2026-04-07
> >
> > Thanks a lot for the reply. My concern still and only remains on the 1d setting of the objective function. It's a rare setting in practice which could not reflect the real case. However, it may be OK for a theory paper to just consider the simple case.
> >
> > I will raise my score to a weak accept.

---

> > > ### Author Response · Authors · 2026-04-07
> > >
> > > We sincerely thank the reviewer for the prompt reply and for raising the score. We genuinely appreciate the open-mindedness and the recognition of the theoretical contribution of our work.
> > >
> > > We agree with the reviewer's intuition that the global landscape of real-world neural networks is vastly more complex than a 1D polynomial. To better bridge our theoretical setting with the "real case", we will explicitly add a discussion in the revised manuscript to clarify how this 1D setting naturally emerges from high-dimensional networks:
> > >
> > > ***The 1D setting as the local equivalent of high-dimensional degeneracy:** Near any local minimum $w^\*$ of a deep network, a Taylor expansion diagonalizes the Hessian, naturally separating the local geometry into strongly convex eigendirections and degenerate eigendirections. Along any degenerate eigendirection $d$ of the Hessian, the restriction of the loss to that direction strictly satisfies:
> > > $$
> > > L(w^\* + \alpha d) = \frac{1}{k}\lambda\_k \alpha^k + O(\alpha^{k+1}),
> > > $$
> > > where $k \ge 4$ is even. This is precisely our prototype $L(x) = \frac{1}{k}x^k$ up to a constant and higher-order terms. While high-dimensional landscapes may also contain mixed-curvature cross-terms, this 1D prototype serves as the exact foundational model for the optimization dynamics along the degenerate subspace (as corroborated by our coupled-mode analysis in Fig. 8). Therefore, the 1D setting is not an isolated toy case, but the necessary building block for understanding Adam's escape from degenerate plateaus in real networks.*
> > >
> > > We hope this addition will make the connection to practice clearer for future readers, and we thank the reviewer for pushing us to articulate this point more explicitly.

---

### Official Review · Reviewer_xfps · 2026-03-13

**Soundness:** 3
**Presentation:** 3
**Significance:** 4
**Originality:** 4
**Overall Recommendation:** 6
**Confidence:** 4

**Summary:**

This paper conducts a very fine-grained analysis of why Adam converges naturally on highly degenerate polynomials without requiring learning rate schedulers. The authors identify a class of objectives $L(x)=\frac{1}{k}x^k$ ($k\geq4$, even) where Adam achieves local linear convergence, fundamentally outperforming the sublinear rates of gradient descent and momentum. The key mechanism is a decoupling between the second moment estimate $v_t$ and the squared gradient $g_t^2$: as gradients vanish, $v_t$ enters autonomous exponential decay, effectively amplifying the learning rate and accelerating convergence from polynomial to exponential. The paper further characterizes a complete hyperparameter phase diagram with three distinct regimes—stable convergence, loss spikes, and SignGD-like oscillation—providing theoretical stability conditions that align closely with empirical results. These findings offer valuable insights into Adam's practical advantages, particularly in deep learning landscapes where degenerate directions are prevalent.

**Compliance With Llm Reviewing Policy:**

Affirmed.

**Final Justification:**

This is an article with obvious merits (please refer to my review for details), and I recommend accepting it.

**Key Questions For Authors:**

Question:

1. The connection between the scalar degenerate polynomial analysis and real neural network training is currently supported by a few preliminary experiments. Would it be possible to measure the coupling ratio $R_t^{(v)}$ along different Hessian eigendirections during actual model training to more directly validate that the decoupling mechanism is indeed responsible for Adam's advantage in practice?

2. The local stability result for full Adam (Theorem 4.1) is local in nature and the basin of attraction is not characterized. Could you provide any rough estimate of how large this basin is, or whether there exist initializations that fail to converge despite the hyperparameters satisfying the stability conditions in Eq. (8) and (9)?

**Limitations:**

Yes

**Strengths And Weaknesses:**

Strengths:

1. Clear and well-motivated problem formulation. The paper identifies a concrete and theoretically clean class of functions (highly degenerate polynomials) that captures an important aspect of real deep learning landscapes, providing a tractable yet meaningful setting for analysis.

2. Strong theory-experiment alignment. The theoretical stability conditions (e.g., $\beta_1 < \beta_2^{k/2(k-2)}$) and predicted convergence rates match empirical results remarkably well, lending high credibility to the analysis.

3. Mechanistic insight is novel and intuitive. The decoupling mechanism between $v_t$ and $g_t^2$, formalized through the coupling ratio $R_t^{(v)}$, provides a clean and compelling explanation for Adam's acceleration that goes beyond prior SignGD-based accounts.

4. Comprehensive phase diagram characterization. The identification of three distinct behavioral regimes (stable convergence, spikes, and SignGD-like oscillation) unifies several previously observed but separately studied phenomena (e.g., loss spikes in Bai et al., 2025) under a single theoretical framework.

Weaknesses:
1. The connection to real deep learning is preliminary. The paper's claim that degenerate directions in neural network landscapes explain Adam's advantage is only supported by a few exploratory experiments (Fig. 15). The gap between isolated degenerate polynomials and coupled, high-dimensional loss surfaces with mixed curvature is not rigorously addressed.

2. Local stability analysis lacks global guarantees for full Adam. While global convergence is established for RMSProp (Theorem 5.7), the full Adam result (Theorem 4.1) is only local. The basin of attraction is not characterized, making it hard to assess how likely the system is to reach the stable fixed point in practice.

Overall, the paper makes solid theoretical contributions with novel mechanistic insights and strong empirical support. This is the kind of paper that, if I were to meet the authors at a conference, I would invite them to give a talk on this work. The weaknesses identified above are relatively minor and do not significantly undermine the core findings. They can reasonably serve as directions for camera-ready revision or future extensions of this work.

---

> ### Author Rebuttal · Authors · 2026-03-28
>
> We sincerely thank the reviewer for the strong support  and high appreciation of our work. Our responses are as follows.
>
> **W1 & Q1: Empirical validation of the decoupling mechanism in real neural networks (e.g., Measuring coupling ratio $R_t^{(v)}$).**
>
> The reviewer's intuition touches upon the very core of our ongoing research—bridging the analytical degenerate models with the exact landscape of real neural networks. We acknowledge that thoroughly disentangling these effects in high-dimensional real networks is highly complex, but we are actively working in this direction. We would like to clarify several key points we have discovered regarding this measurement, which will form the basis of our future comprehensive empirical study:
>
> **(1) The Universality of Decoupling (Both Degenerate & Strongly Convex):** The decoupling between $v_t$ and $g_t^2$ occurs generically, not just in degenerate directions, but also in sharp/flat quadratic directions. As illustrated in Appendix (e.g., Fig. 9) for quadratic cases, $g_t^2$ decays rapidly, leading to the decoupling of $v_t$. Our theoretical result (Lem. F.1) reveals a fascinating dynamic: for RMSProp, this decoupling-induced learning rate amplification leads to super-linear convergence in quadratic settings. However, for full Adam, if $g_t$ decays too quickly (exceeding the exponential rate $\beta_1$), the momentum $m_t$ becomes bottlenecked by its own memory. Thus, the initial convergence of Adam on quadratics remains linear (though it eventually spikes). Therefore, $R_t^{(v)}$ can be massive in both degenerate and non-degenerate directions, a phenomenon we have indeed observed in specific parameter blocks during actual Transformer training.
>
> **(2) The Geometric Distinction: True Degeneracy vs. Small Eigenvalues:** Identifying true degenerate directions requires rigorous geometric probing beyond simply finding "small eigenvalues". If an eigenvalue is merely small but strictly positive (a flat quadratic valley), GD still enjoys linear convergence. However, on a true degenerate direction, GD suffers from power-law (sub-linear) decay. To isolate this, our ongoing empirical strategy involves probing the local geometry around a converged minimum $w^*$. By moving along a near-zero eigenvector $d_{zero}$ via $w^\* + \alpha d_{zero}$, we compute the exact local curvature using Hessian-Vector Products (HVP): $h(\alpha) = d_{zero}^\top \nabla^2 L(w^\* + \alpha d_{zero}) d_{zero}$. If it is a flat quadratic valley, $h(\alpha)$ remains a non-zero constant $\lambda$ regardless of $\alpha$. If it is a true 4th-order degeneracy ($\nabla^2 L \propto x^2$), then $h(\alpha) \propto \alpha^2$, meaning the curvature itself vanishes as $\alpha \to 0$. This HVP probing is essential to mathematically isolate where Adam's order-of-magnitude acceleration genuinely occurs.
>
> **(3) Disentangling the Sources of Acceleration:** Ultimately, Adam accelerates training in non-degenerate directions as well, but fundamentally differently. In strongly convex (quadratic) directions, Adam provides a constant-factor improvement in the linear convergence rate (e.g., in Fig. 1(a), both GD and Adam converge exponentially, but Adam has a strictly steeper slope). In highly degenerate directions, Adam provides an order-of-complexity improvement (shifting from sub-linear to linear). Completely disentangling these two entangled acceleration effects in a dynamic, high-dimensional trajectory requires extremely delicate layer-wise and phase-wise spectral tracking, which is the primary focus of our immediate future work.
>
> **W2 & Q2: Characterizing the Basin of Attraction for full Adam (Thm. 4.1).**
>
> We acknowledge that characterizing the exact global convergence boundary for Thm. 4.1 is challenging due to strong nonlinear coupling. We will add a “Remark on the Basin of Attraction” to clarify this from both analytical and empirical perspectives:
>
> **(1) Theoretical Complexity of Full Adam:** From the perspective of nonlinear dynamics, deriving a closed-form global boundary for full Adam is highly intractable. In the case of RMSProp, the state space dimensionality is reduced, allowing us to leverage the monotonicity of the discrete mapping (e.g., Lem. B.1, B.3) and construct a global proof using Coppel's (1955) convergence theory. However, the introduction of momentum $m_t$ creates a strong nonlinear coupling between $\omega_t$ and $\lambda_t$. The system evolves into a 3-dimensional discrete dynamical system, whose basin may exhibit complex, initialization-dependent geometry over $(x_0,m_0,v_0)$.
>
> **(2) Empirical Basin under Standard Initialization:** Although the exact boundary is hard to characterize, empirical evidence shows a broad and robust effective basin. In Fig. 3(b), under standard initializations ($m_0 = g_0$, $v_0=g_0^2$), the observed convergence region closely matches the phase diagram predicted by local stability. Thus, typical trajectories reliably enter the linear convergence basin of Thm. 4.1.

---

> > ### Author Rebuttal · Reviewer_xfps · 2026-04-02
> >
> > Thanks for the author's reply. I will keep my score. I recommend accepting this article. The conclusion of this article is highly inspiring for my own theory result.

---

> > > ### Author Response · Authors · 2026-04-08
> > >
> > > We sincerely thank the reviewers for their exceptionally positive evaluation.

---

### Official Review · Reviewer_x93G · 2026-03-16

**Soundness:** 3
**Presentation:** 3
**Significance:** 3
**Originality:** 3
**Overall Recommendation:** 4
**Confidence:** 3

**Summary:**

An interesting and original paper. It argues that Adam can have a true asymptotic advantage on highly degenerate polynomial objectives: Adam achieves stable linear convergence, while GD and momentum remain sublinear. The key idea is that Adam’s second-moment term can decouple from the current gradient scale, creating an implicit acceleration mechanism. The theory is clean and the experiments appear consistent with it.

**Compliance With Llm Reviewing Policy:**

Affirmed.

**Key Questions For Authors:**

The insights from the paper are interesting. Have the authors gave a thought about how could the acceleration results tried to tie the observations in neural net training that Adam could be faster? And how does the intuition provided in the paper compared with related works analyzing advantages of Adam compared with SGD?

**Limitations:**

yes.

**Strengths And Weaknesses:**

Strengths:
1. Clear theoretical gap between methods. The paper identifies a concrete class of problems where Adam/RMSProp are provably faster than GD and momentum, which makes the result feel substantive.
2. The phase diagram is interesting. The stable, spike, and oscillatory regimes are tied to fixed-point behavior, which gives a simple way to think about different Adam dynamics.
3. The experiments support the theory reasonably well. The paper checks the predicted phase boundaries and convergence behavior.

Weaknesses:
1. The theory is built on a narrow setting. Most of the main results are for highly degenerate polynomial objectives, so it is hard to know how far they carry beyond that setting.

---

> ### Author Rebuttal · Authors · 2026-03-28
>
> We sincerely thank the reviewer for high appreciation of our work. Below, we provide point-by-point clarifications.
>
> **W1: The theory is built on a narrow setting.**
>
> We address this from three angles:
>
> **(1) Theoretical Necessity of Isolation.** Near any local minimum, a Taylor expansion decomposes the loss into strongly convex and degenerate directions; $L(x) = \frac{1}{k}x^k$ precisely captures the latter. This isolation lets us rigorously establish the $v_t$–$g_t^2$ decoupling mechanism, derive tight stability conditions, and obtain the explicit convergence rate $\beta_2^{1/(2(k-2))}$ — all with strong empirical alignment.
>
> **(2) Results Extend to a Broader Function Class.** Thm. 4.1 extends directly to $L(x) = \frac{1}{k}x^k(1+h(x)) (k\geq 4)$ where $h$ is analytic and $h(0)=0$. Since $g(x) = \Theta(x^{k-1})$ as $x\to 0$, the higher-order terms from $h$ vanish when evaluating the Jacobian at $(\omega^\*, \lambda^\*, 0)$. The stability conditions and convergence rate are therefore **identical** for this entire function class, meaning our results characterize the universal behavior of any loss with a $k$-th order degenerate minimum. We will incorporate this as a formal remark in the revision.
>
> **(3) High-Dimensional Mixed-Curvature Analysis.** Section 7 and Appendix F analyzed the mixed-curvature and quadratic cases. In landscapes coupling quadratic and degenerate directions, the optimization bottleneck is determined by the flattest directions — precisely those our model captures. Hessian spectral analysis of real networks (Fig. 15(c,f)) consistently shows eigenvalues concentrated near zero, supporting the prevalence of degenerate directions in practice. We acknowledge this falls short of a formal proof for the general case and will add an explicit limitations discussion.
>
> ---
>
> **Q1: Connection between theoretical acceleration and practical neural network training.**
>
> We establish this connection through a logical chain:
>
> **Steps 1–3: The Core Mechanism.** As $x_t \to 0$ along degenerate directions, the gradient $g_t$ vanishes faster than the memory decay rate of $v_t$, causing $v_t$ to decouple from $g_t^2$ and enter autonomous exponential decay ($v_t \approx \beta_2 v_{t-1}$). This induces an exponentially growing effective learning rate $\eta_{\text{eff},t} \propto \beta_2^{-t/2}$, converting sublinear into linear convergence (Lem. 5.4–5.5, Thm. 5.7). GD and Momentum, lacking this mechanism, remain trapped in the $\Theta(t^{-1/(k-2)})$ curse of degeneracy (Thm. 5.1 and 5.3).
>
> **Step 4: Neural Networks Are Intrinsically Degenerate.** This mechanism is directly relevant in practice because deep learning loss landscapes are pervasively degenerate. Hessian analysis reveals eigenvalues concentrated near zero (Sagun et al., 2017), and this is structural rather than incidental. The Embedding Principle (Zhang et al., 2021; 2022b; Bai et al., 2024) explains why: critical points of small networks embed as high-dimensional degenerate manifolds in larger networks. High-order degeneracy is thus a fundamental topological feature of deep network landscapes.
>
> **Step 5: Empirical Evidence.** Section 7 and Figure 15 provide initial support. Replacing ReLU with Softmax in MLPs increases landscape degeneracy — the Hessian spectral density concentrates more near zero (Fig. 15(c)) — and Adam's advantage over SGD grows correspondingly (Fig. 15(b) vs. 15(a)). Transformers similarly exhibit higher degeneracy than CNNs (Fig. 15(f)), consistent with Adam's well-documented advantage in language model. This points to a concrete mechanistic pathway: architectures with more degenerate directions suffer more from the curse of degeneracy under GD, while Adam's decoupling mechanism provides proportionally greater acceleration.
>
> ---
>
> **Q2: Comparison with related works on Adam's advantages over SGD.**
>
> Prior mechanisms each rely on specific structural assumptions. The table below summarizes key distinctions:
>
> | Work | Proposed Mechanism | Key Assumption | Requires Stochasticity |
> |---|---|---|---|
> | Zhang et al. (2020) | Heavy-tailed noise | Heavy-tailed gradient noise | Yes |
> | Zhang et al. (2024) | Hessian block heterogeneity | Block-diagonal Hessian structure | No |
> | Tomihari & Sato (2025) | Gradient heterogeneity | Heterogeneous gradients and SignGD approximation | No |
> | Xie et al. (2024) | $\ell_\infty$ geometry exploitation | Coordinate-wisely smooth w.r.t. $\ell_\infty$ norm | No |
> | Kunstner et al. (2024) | Heavy-tailed class imbalance | Imbalanced data distribution| No |
> | **Ours** | $v_t$–$g_t^2$ decoupling on degenerate directions | **Loss landscape degeneracy** | No |
>
> Our mechanism requires only that degenerate directions exist — a structural property that holds universally in deep networks, independent of data distribution, Hessian block structure, or noise. These mechanisms are not mutually exclusive; multiple factors likely operate simultaneously in real training, and our work isolates one that has been overlooked.

---

> > ### Author Rebuttal · Reviewer_x93G · 2026-04-07
> >
> > I appreciate the authors' response, my questions are addressed and I will keep my score

---

> > > ### Author Response · Authors · 2026-04-08
> > >
> > > We thank the reviewer for their positive score and constructive feedback, as well as their continued engagement with our manuscript.

---

### Decision · Program_Chairs · 2026-04-30

**Decision:**

Accept (spotlight)

**Comment:**

This paper addresses an important and largely unresolved question: under what classes of objective functions does Adam exhibit inherent advantages. To this end, the authors introduce a clear and well-defined function class, highly degenerate polynomials, and provide a detailed analysis of the resulting convergence behavior. In particular, the paper identifies a decoupling mechanism between the second-moment estimate and the squared gradient, showing that this leads to an effective learning rate that grows over time. As a consequence, while Gradient Descent and Momentum exhibit sublinear convergence, Adam is shown to achieve linear convergence in this setting. In addition, the paper presents a phase diagram that unifies the behavior of Adam across hyperparameter regimes (stable convergence, spikes, and SignGD-like oscillations), offering a new perspective on the dynamics of adaptive optimization methods.

The reviewers’ evaluations are overall positive, with consistent appreciation for the clarity of the theory, the novelty of the mechanistic insight, and the strong alignment between theoretical predictions and empirical results. In particular, the paper’s ability to quantitatively demonstrate Adam’s advantage on a concrete class of problems, together with the proposed decoupling-based acceleration mechanism, is regarded as an important contribution that could serve as a foundation for future theoretical work.

While several concerns were raised, the authors provided thorough and satisfactory responses in the rebuttal, and most reviewers indicated that their concerns were adequately addressed.

Overall, although the considered setting has certain limitations, the paper offers clear and convincing theoretical results within this scope, along with a novel mechanistic understanding. It therefore constitutes a meaningful contribution to both optimization theory and the theoretical understanding of deep learning.